# CX-5461 activates the DNA damage response and demonstrates therapeutic efficacy in high-grade serous ovarian cancer

Elaine Sanij [1,2,3,18✉], Katherine M. Hannan [4,5,18✉], Jiachen Xuan[1,3], Shunfei Yan[1,3], Jessica E. Ahern[1], Anna S. Trigos [1,3], Natalie Brajanovski[1], Jinbae Son[1,3], Keefe T. Chan [1], Olga Kondrashova [6,7], Elizabeth Lieschke[6], Matthew J. Wakefield [6,8], Daniel Frank[6,7], Sarah Ellis [1,3], Carleen Cullinane [1,3], Jian Kang[1], Gretchen Poortinga [1,3,9], Purba Nag [10,11], Andrew J. Deans [9,12], Kum Kum Khanna[10], Linda Mileshkin[1,3], Grant A. McArthur[1,2,3,9], John Soong[13], Els M. J. J. Berns[14], Ross D. Hannan [1,3,4,5,15,16], Clare L. Scott [1,6,17], Karen E. Sheppard [1,3,5,19] & Richard B. Pearson [1,3,5,15,19✉]

Acquired resistance to PARP inhibitors (PARPi) is a major challenge for the clinical management of high grade serous ovarian cancer (HGSOC). Here, we demonstrate CX-5461, the first-in-class inhibitor of RNA polymerase I transcription of ribosomal RNA genes (rDNA), induces replication stress and activates the DNA damage response. CX-5461 co-operates with PARPi in exacerbating replication stress and enhances therapeutic efficacy against homologous recombination (HR) DNA repair-deficient HGSOC-patient-derived xenograft (PDX) in vivo. We demonstrate CX-5461 has a different sensitivity spectrum to PARPi involving MRE11-dependent degradation of replication forks. Importantly, CX-5461 exhibits in vivo single agent efficacy in a HGSOC-PDX with reduced sensitivity to PARPi by overcoming replication fork protection. Further, we identify CX-5461-sensitivity gene expression signatures in primary and relapsed HGSOC. We propose CX-5461 is a promising therapy in combination with PARPi in HR-deficient HGSOC and also as a single agent for the treatment of relapsed disease.

[1] Peter MacCallum Cancer Centre, 305 Grattan St, Melbourne, VIC 3000, Australia. [2] Department of Clinical Pathology, University of Melbourne, Parkville, VIC 3010, Australia. [3] Sir Peter MacCallum Department of Oncology, University of Melbourne, Parkville, VIC 3010, Australia. [4] ACRF Department of Cancer Biology and Therapeutics, The John Curtin School of Medical Research, Australian National University, Acton 2601 Australia Capital Territory, Australia. [5] Department of Biochemistry and Molecular Biology, University of Melbourne, Parkville, VIC 3010, Australia. [6] The Walter and Eliza Hall Institute of Medical Research, Parkville, VIC 3052, Australia. [7] Department of Medical Biology, University of Melbourne, Parkville, VIC 3010, Australia. [8] Melbourne Bioinformatics, University of Melbourne, Victoria 3010, Australia. [9] Department of Medicine, St. Vincent's Hospital, University of Melbourne, Parkville, VIC 3010, Australia. [10] QIMR Berghofer Medical Research Institute, Brisbane, QLD, Australia. [11] School of Environment and Sciences, Griffith University, Nathan, Brisbane, QLD 4111, Australia. [12] Genome Stability Unit, St Vincent's Institute, Fitzroy, VIC 3065, Australia. [13] Senhwa Biosciences, Virginia Commonwealth University School of Medicine, San Diego, CA, USA. [14] Department of Medical Oncology, Erasmus MC Cancer Institute, Rotterdam, The Netherlands. [15] Department of Biochemistry and Molecular Biology, Monash University, Clayton, VIC 3800, Australia. [16] School of Biomedical Sciences, University of Queensland, Brisbane, QLD 4072, Australia. [17] Department of Medicine and Health Sciences, Monash University, Clayton, VIC 3168, Australia. [18] These authors contributed equally: Elaine Sanij, Katherine M. Hannan. [19] These authors jointly supervised this work: Karen E. Sheppard, Richard B. Pearson. ✉email: elaine.sanij@petermac.org; kate.hannan@anu.edu.au; rick.pearson@petermac.org

Ovarian cancer (OVCA) is the major cause of death from gynaecological cancers. The high-grade serous ovarian cancer (HGSOC) subtype accounts for 70–80% of OVCA deaths and overall survival has not changed for several decades[1]. HGSOC is almost invariably *TP53*-mutant with 50% of HGSOC demonstrating deficiencies in homologous recombination (HR) DNA repair of DNA double strand breaks (DSB), most commonly due to mutations in *BRCA1/2*[2]. Tumours with HR deficiency (HRD) exhibit favourable responses to chemotherapy and Poly-(ADP-ribose) polymerase (PARP) inhibitors (PARPi)[2–4]. PARP enzymes are involved in DNA repair through activation of the base excision repair (BER) and alternative end-joining pathways and inhibition of the non-homologous end-joining (NHEJ) pathway[4]. The sensitivity of HR-deficient cells to PARPi relies on overuse of the NHEJ pathway, impaired DNA replication fork protection and persistence of unrepaired collapsed forks[4–6].

PARPi are now utilized as maintenance therapy following complete or partial response to platinum-based chemotherapy in recurrent HGSOC[7]. More recently, PARPi have shown substantial benefit with regard to progression-free survival among women with newly diagnosed advanced OVCA with *BRCA1/2* mutations[8]. However, resistance to PARPi has been associated with multiple mechanisms including secondary mutations in genes involved in the HR pathway and stabilization of DNA replication forks[9–11]. Thus, the development of strategies to overcome resistance to PARPi will provide a significant advancement in the treatment of HGSOC.

Hyperactivation of RNA polymerase I (Pol I) transcription of the 300 copies of ribosomal RNA (rRNA) genes (rDNA) is a consistent feature of cancer cells[12]. The rDNA repeats are transcribed to produce the 47S pre-rRNA, containing the sequences of the 18S, 5.8S and 28S rRNA components of the ribosome. We have demonstrated targeting Pol I transcription using the small-molecule inhibitor CX-5461 is an exciting approach for cancer treatment[13–15]. The first-in-human trial of CX-5461 in patients with advanced haematological cancers (Peter MacCallum Cancer Centre) has demonstrated single-agent anti-tumour activity in *TP53* wild type and *TP53*-mutant haematologic malignancies[16]. CX-5461 is also in phase I clinical trial in solid tumours (Canadian Cancer Trials Group, NCT02719977)[17].

We and others have demonstrated that CX-5461 activates a p53-independent DNA damage response (DDR) leading to S-phase delay and G2 cell cycle arrest[17–20]. We have shown CX-5461 induces ATM (acute Ataxia telangiectasia mutated) and ATR (Ataxia telangiectasia and Rad3) kinase signalling in primary fibroblasts prior to the detection of indicators of DNA damage across the genome[19]. We also showed that CX-5461 in combination with dual inhibition of checkpoint kinases 1/2 (CHK1/2) downstream of ATM and ATR signalling significantly enhanced the therapeutic outcome of p53-null MYC-driven lymphoma in vivo[19]. More recently, CX-5461 was shown to exhibit synthetic lethality with *BRCA1/2* deficiency[17]. Chronic treatment with CX-5461 in HCT116 colon carcinoma cells was reported to induce stabilization of G-quadruplex DNA (GQ) structures, leading to defects in DNA replication, which presumably require the HR pathway to resolve these defects. However, CX-5461 demonstrated a different spectrum of cytotoxicity compared with the PARPi olaparib across breast cancer cell lines[17]. This suggests that additional mechanisms to HR defects underlie sensitivity to CX-5461. Recently, the sensitivity profile of CX-5461 was shown to closely resemble a topoisomerase II (TOP2) poison[21,22]. TOP2a is an essential component of the Pol I pre-initiation complex[23] and while CX-5461 demonstrates highly selective inhibition of Pol I transcription initiation, it is plausible that it does so by trapping TOP2 at rDNA and potentially across the genome.

In this report, we demonstrate that sensitivity to CX-5461 is associated with BRCA mutation and MYC targets gene expression signatures. We show CX-5461 activates ATM/ATR signalling and a G2/M cell cycle checkpoint in HR-proficient HGSOC cells but it induces cell death in HR-deficient HGSOC. Mechanistically, we show that CX-5461 activates ATR and this is associated with replication stress and does not involve stabilization of GQ structures as previously proposed. CX-5461 activation of ATR is associated with global replication stress and DNA damage involving MRE11-dependent degradation of DNA replication forks. We demonstrate that as single agents CX-5461 and PARPi exhibit different mechanisms of destabilizing replication forks. Importantly, the combination of CX-5461 and PARPi leads to exacerbated replication stress, DNA damage, pronounced cell cycle arrest and inhibition of clonogenic survival of HR-proficient HGSOC cells and exhibits greater efficacy in HR-deficient HGSOC cells. Thus, our data unveil a CX-5461/ PARPi and HRD synthetic lethality axis. Furthermore, the combination of CX-5461 and PARPi leads to significantly improved regression of HR-deficient HGSOC-PDX tumours in vivo. Importantly, we also provide evidence that CX-5461 has significant in vivo therapeutic benefit in HGSOC-PDX with reduced sensitivity to olaparib by overcoming fork protection, a common PARPi resistance mechanism. Here, we also identify predictive signatures of CX-5461 sensitivity in primary and relapsed OVCA samples highlightling the potential of CX-5461 therapy in primary, chemotherapy- and PARPi-resistant HGSOC.

## Results

**Activity of CX-5461 in OVCA cell lines**. The in vitro effects of CX-5461 on human OVCA cells were evaluated using a panel of 32 established human OVCA cell lines. These cell lines were selected to be representative of a range of histologic OVCA subtypes (Supplementary Table 1). Increasing concentrations (1 - nM–10 μM) of CX-5461 were used to assess the concentration of drug that induced a 50% reduction in cell proliferation ($GI_{50}$) at 48 hours (h). The $GI_{50}$ values varied between individual cell lines and ranged from 12 nM for OVCAR3 to 5.17 μM for OV90 (Fig. 1a). The cell lines were defined as sensitive to CX-5461 if the $GI_{50}$ was below the geometric median of 363 nM. There was no statistically significant correlation between *TP53* mutation status and sensitivity to CX-5461 (Supplementary Table 1, Fig. 1b). The efficacy of growth inhibition by CX-5461 correlated with a higher rate of basal rDNA transcription in the sensitive compared with the resistant OVCA cell lines (Fig. 1c). However, both CX-5461 sensitive and resistant cell lines displayed similar levels of inhibition of Pol I transcription following 1 h treatment with CX-5461, with concentrations that inhibit Pol I transcription by 50% ($IC_{50}$) ranging between 38 and 285 nM (Fig. 1d and e). The data shows CX-5461 is on-target in inhibiting Pol I transcription at doses 10-fold less than the plasma concentrations range (584.1 nM–3.3 μM) used in the Phase I CX-5461 dose escalation study[16].

**CX-5461 exhibits synthetic lethality with HRD in HGSOC**. To interrogate the molecular mechanisms underlying CX-5461 sensitivity in OVCA, gene expression profiles for 12 CX-5461-sensitive and 11 -resistant cell lines were generated (Fig. 1a). Gene set enrichment analysis (GSEA) identified BRCA1 mutation and MYC targets gene expression signatures to correlate with sensitivity to CX-5461 in vitro (Fig. 2a, Supplementary Fig. 1). Indeed, we observed a significant enrichment of a HRD gene expression signature[24] in the CX-5461-sensitive cell lines (Fig. 2b). However, correlation of CX-5461 and PARPi sensitivity across the HGSOC cell lines was not evident (Supplementary Fig. 2), indicating that additional mechanisms to HRD confer sensitivity to CX-5461.

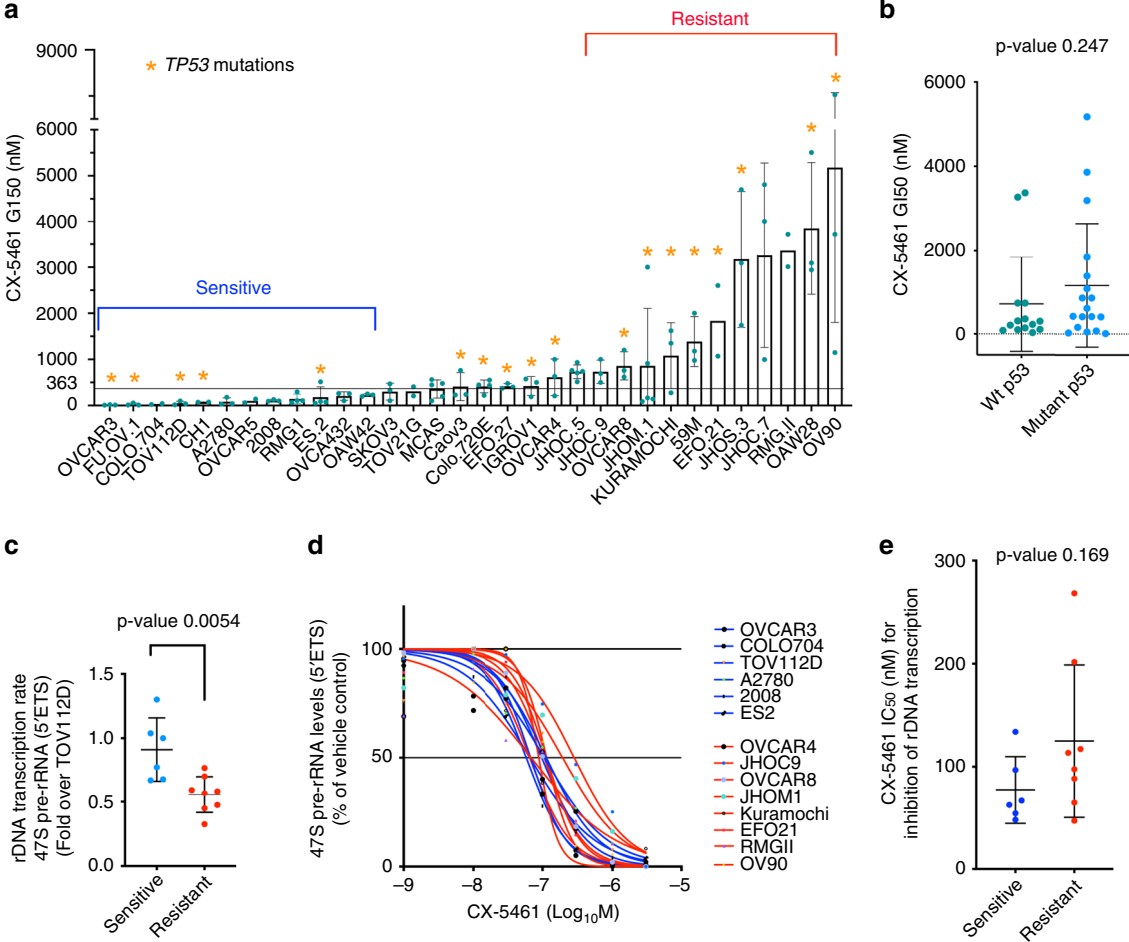

**Fig. 1 Sensitivity of ovarian cancer (OVCA) cell lines to CX-5461. a** CX-5461 doses that induce 50% growth inhibition (GI$_{50}$) at 48 hours. Error bars represent mean ± standard deviation (SD). Asterisks denote mutated *TP53* status. The geometric mean GI$_{50}$ dose of 363 nM is indicated by the fine line. Information regarding each cell lines source, mean G1$_{50}$ values, SD and N values are outlined in Supplementary Table 1. The 12 CX-5461-sensitive and 11 CX-5461-resistant cell lines highlighted were utilised to identify CX-5461-sensitivity gene expression signatures in Fig. 2a. **b** CX-5461 GI$_{50}$ doses of *TP53* wild type and *TP53* mutant OVCA cell lines as outlined in (**a**). Statistical analysis was performed using a two-sided Mann–Whitney test. Error bars represent mean ± SD. **c** Baseline rDNA transcription rates of OVCA cell lines (listed in (**d**)), as determined by quantitative real-time PCR (qRT-PCR) analysis using primers specific to the external transcribed spacer 5'ETS (+413–521 bp) relative to the transcription start site (TSS). Expression levels in each cell line were normalised to Vimentin mRNA and expressed as fold change relative to TOV112D cells. Each dot represents the mean value of $n = 3$ biologically independent experiments per cell line (Individual data points are provided in Supplementary Data 4). Error bars represent mean ± SD. Statistical analysis was performed using a two-tailed unpaired t test. **d** Dose-response curves of 47S precursor rRNA levels measured by quantitative RT-PCR. OVCA cell lines were treated with vehicle or increasing drug concentrations of CX-5461 (0.001, 0.01, 0.03, 0.1, 0.3, 1 and 3 μM) for 1 h before RNA was harvested. Expression levels were normalised to Vimentin mRNA and expressed as fold change relative to vehicle-treated controls. Shown are the mean of $n = 3$ independent biological experiments for each cell line. (Individual data points are provided in Supplementary Data 4). **e** The mean inhibitory concentration values that led to 50% transcription inhibition (IC$_{50}$) (of $n = 3$ biologically independent experiments for each cell line) from D are presented (Individual data points are provided in Supplementary Data 4). Error bars represent mean ± SD. Statistical analysis was performed using a two-sided Mann–Whitney test.

The level of the BRCA1 mutation (BRCAm-sig) and oncogenic MYC (MYC_UP-sig) gene expression signatures accurately distinguished the CX-5461-sensitive and -resistant cell lines in an independent dataset from the Broad Institute Cancer Cell Line Encyclopaedia (CCLE) (Fig. 2c). We found that using both the BRCAm-sig and MYC_UP-sig gene expression signatures provided greater power for predicting sensitivity to CX-5461 than using the BRCAm-sig as sole predictor (ANOVA Chi-squared test *p*-value = 0.026557). Indeed, sensitivity to CX-5461 was also associated with high basal rates of Pol I transcription (Fig. 1c). As high MYC activity is known to drive Pol I transcription at multiple levels[25] the data is consistent with elevated MYC activity driving sensitivity to CX-5461. Taken together, our data suggest CX-5461 may provide therapeutic benefit in the high-MYCN

HGSOC subtype classified with elevated functional MYC activity and poor progression-free survival[26,27].

To confirm CX-5461 synthetic lethal interaction with HRD, we tested the effects of short interfering RNAs (siRNAs) targeting HR genes on the efficacy of CX-5461 and demonstrated strong synergistic inhibition in HGSOC OVCAR4 cells, reported to closely resemble the genomic profile of HGSOC tumours[28] (Fig. 2d). Further, we tested the growth inhibitory effects of CX-5461 in isogenically matched HR-proficient OVCAR8 cells and a derivative HR-deficient RAD51C knockout (KO) OVCAR8 cell line previously characterized to lack the capacity to form RAD51 foci in response to ionizing radiation (IR)[29] (Supplementary Fig. 3a–c. The HR-deficient cell line exhibited increased sensitivity to growth inhibition by CX-5461 at the low doses of 10

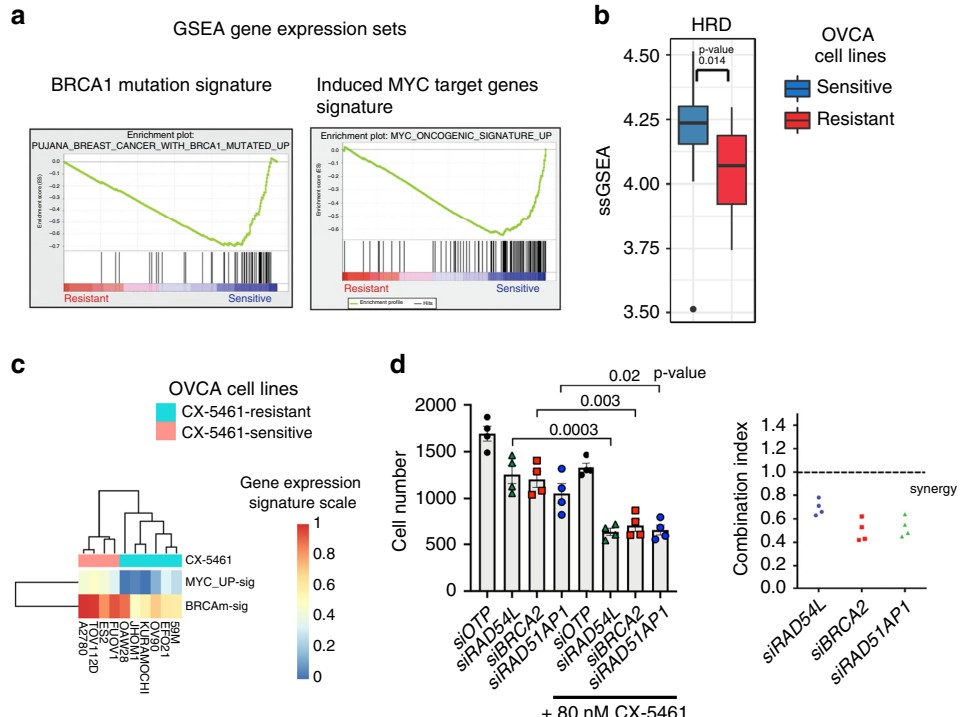

**Fig. 2 Sensitivity to CX-5461 is associated with induced MYC target genes, BRCA1-mutated and HRD gene expression signatures. a** GSEA of microarray expression data of 12 CX-5461-sensitive and 11 CX-5461-resistant cell lines (Fig. 1a). Enrichment plots of gene sets identified to be enriched in the CX-5461-sensitive cell lines are shown. **b** Single sample GSEA (ssGSEA) was utilized to obtain the level of activity of a HRD gene expression signature[24] in individual samples. Genes in each sample were ranked according to their expression levels, and a score for each pathway was generated based on the empirical cumulative distribution function, reflecting how highly or lowly genes were found in the ranked list. $n = 32$ cell lines in Fig. 1a, box plot—median, upper and lower hinges correspond to the first and third quartiles (25th and 75th percentiles). Upper whisker extends to the largest value no further than 1.5*IQR (IQR = inter-quartile range). Lower whisker extends to the smallest value at 1.5*IQR. Data beyond the end of the whiskers are plotted as individual points. Statistical significance was obtained using two-sided Wilcoxon tests. **c** The level of expression of MYC target genes (MYC_UP-sig) and BRCA1 mutated (BRCAm-sig) gene signatures were calculated in RNA expression data from the Broad Institute CCLE using ssGSEA. The MYC_UP-sig and BRCAm-sig gene expression signatures are more highly expressed in the CX-5461-sensitive group compared with the resistant group (one-sided Wilcoxon tests $p$-value 0.004762 and 0.009524, respectively). **d** The combination of CX-5461 and siRNAs targeting HR genes in OVCAR4 cells synergistically inhibits proliferation. Four individual siRNA duplexes per gene were reversed transfected for 24 h, followed by treatment with CX-5461 (80 nM) or vehicle for 48 h. Cell counts were measured using DAPI staining and imaged using Cellomics. $n = 4$, each data point represents individual siRNA duplexes. Error bars represent mean ± SEM (standard error of the mean). Statistical analysis was performed using a two-sided one-way ANOVA, Tukey's multiple comparisons test (adjusted $p$-values are shown). Accompanying graph (right) is a bliss plot with each dot representing an individual siRNA duplex. A combination index of CI < 1 indicates synergy, CI > 1 indicates antagonism and CI = 1 indicates additive effect.

and 100 nM compared with OVCAR8 cells (Fig. 3a). In HR-proficient OVCAR8 cells, CX-5461-mediated inhibition of proliferation was associated with a pronounced G2/M cell cycle arrest measured by BrdU/PI staining (Fig. 3b and Supplementary Fig. 3D) and utilising the FUCCI reporter system (Fig. 3c and Supplementary Fig. 3E). OVCAR8 cells treated with CX-5461 also exhibited cytokinesis failure and multinucleation (8 N DNA content) (Fig. 3d and Supplementary Fig. 3D) suggesting that CX-5461 induces defects in chromosome segregation possibly due to persistent DNA replication stress and DNA damage in mitosis[30]. In comparison, HR-deficient RAD51C KO OVCAR8 cells underwent cell death at 100 nM and 1 µM CX-5461 further confirming synthetic lethal CX-5461/HRD interaction (Fig. 3d and Supplementary Fig. 3D).

Next, we investigated CX-5461 mediated activation of ATM/ATR signalling in HGSOC cell lines (Fig. 3e). In agreement with our previous findings[19], CX-5461 induced ATM/ATR signalling, as indicated by increased phosphorylation of ATR T1989, ATM S1981, CHK1 S345, CHK2 T68 and low induction in γH2AX levels in both cell lines. Greater increases in S4/S8 phosphorylation of RPA32 (replication protein A), which protects single-strand DNA (ssDNA), were observed in HR-deficient cells

following 100 nM and 1 µM CX-5461 compared with HR proficient OVCAR8 cells indicating persistent stalled replication forks[31].Thus, CX-5461 induces replication stress in HR-proficient cells associated with increased percentage of multinucleated cells while HR-deficient cells undergo cell death following CX-5461 treatment due to exacerbated replication stress. Altogether, these findings demonstrate that CX-5461 exhibits strong antiproliferative effects in OVCA cells and that HR defects sensitize HGSOC cells to CX-5461-mediated cell death, consistent with the BRCA1-mutated and HRD gene expression signatures predicting OVCA cell sensitivity to CX-5461 (Fig. 2a, b).

**CX-5461 induces replication stress and DDR.** To examine the mechanism of activation of ATM/ATR signalling, we investigated the effects of CX-5461 treatment on the formation of GQ structures in OVCAR8, OVCAR8 RAD51C KO and OVCAR4 cells (Fig. 4a, b and Supplementary Fig. 4A–D). CX-5461 did not induce GQ stabilization in HR-proficient OVCAR8 or OVCAR4 cells nor in HR-deficient OVCAR8 RAD51C KO cells compared with a bona fide GQ stabilizer (TMPyP4) at 1 h, 3 h and 24 h post-treatment. Intriguingly, GQ staining was significantly decreased in OVCAR8 cells treated with CX-5461 for 3 h.

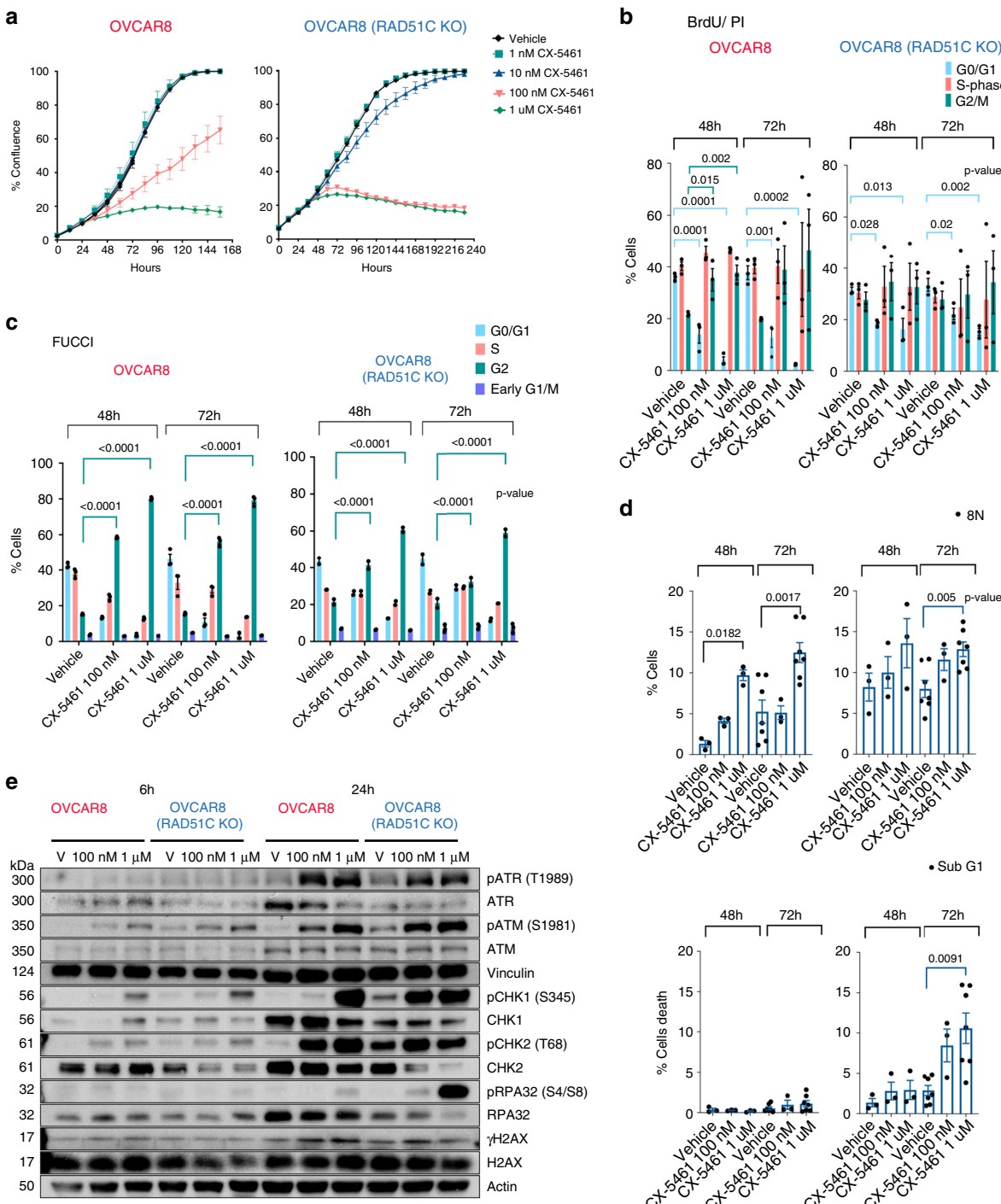

**Fig. 3 CX-5461 is synthetic lethal with HRD in HGSOC. a** In vitro CX-5461 dose response proliferation time course, assessed by cell confluency using IncuCyte ZOOM of OVCAR8 and OVCAR8 RAD51C KO cell lines. Representative of two biologically independent experiments. Error bars represent mean ± SEM of $n = 5$ technical replicates. **b** Cell cycle analysis of cells treated with vehicle, 100 nM or 1 μM CX-5461 for 48 hours and 72 h and labelled with BrdU for 30 min prior to harvest. Analytical FACS analysis of BrdU incorporation as a function of DNA content was performed (Representative plots and gating strategy are shown in Supplementary Fig. 3D). Histogram plots displaying the percentage of G0/G1 (blue) and G2M (green) and S-phase BrdU-labelled (red) cell populations. Error bars represent mean ± SEM of $n = 3$ biologically independent experiments. **c** Quantitation of cell cycle profiles using FUCCI-labelled cells treated with vehicle, 100 nM or 1 μM CX-5461 for 48 and 72 h. Representative flow cytometry profiles and gating strategy are shown in Supplementary Fig. 3E. Error bars represent mean ± SEM of $n = 3$ biologically independent experiments. **d** Histogram plots displaying the percentage of cells with >4n DNA content (top panel) and Sub G0/G1 cell populations (bottom panel) as detected by BrdU/PI cell cycle analysis described in (**b**) and Supplementary Fig. 3D, ($n = 3$ biologically independent experiments, $n = 7$ for OVCAR8 and OVCA8 RAD51C KO cells treated with vehicle or 1 μM CX-5461 for 72 h), error bars represent mean ± SEM. Statistical analysis in B-D was performed using a two-sided one-way ANOVA, Tukey's multiple comparisons test (adjusted $p$-values are shown). **e** Western blot analysis of cells treated with either vehicle, 100 nM or 1 μM CX-5461 for 6 and 24 h. Representative of $n = 3$ biologically independent experiments. Blots shown are of samples derived from the same experiment and were processed in parallel. Loading controls Vinculin and Actin were processed by re-probing the blots. Full sized scan of western blots are provided in Supplementary Fig. 10.

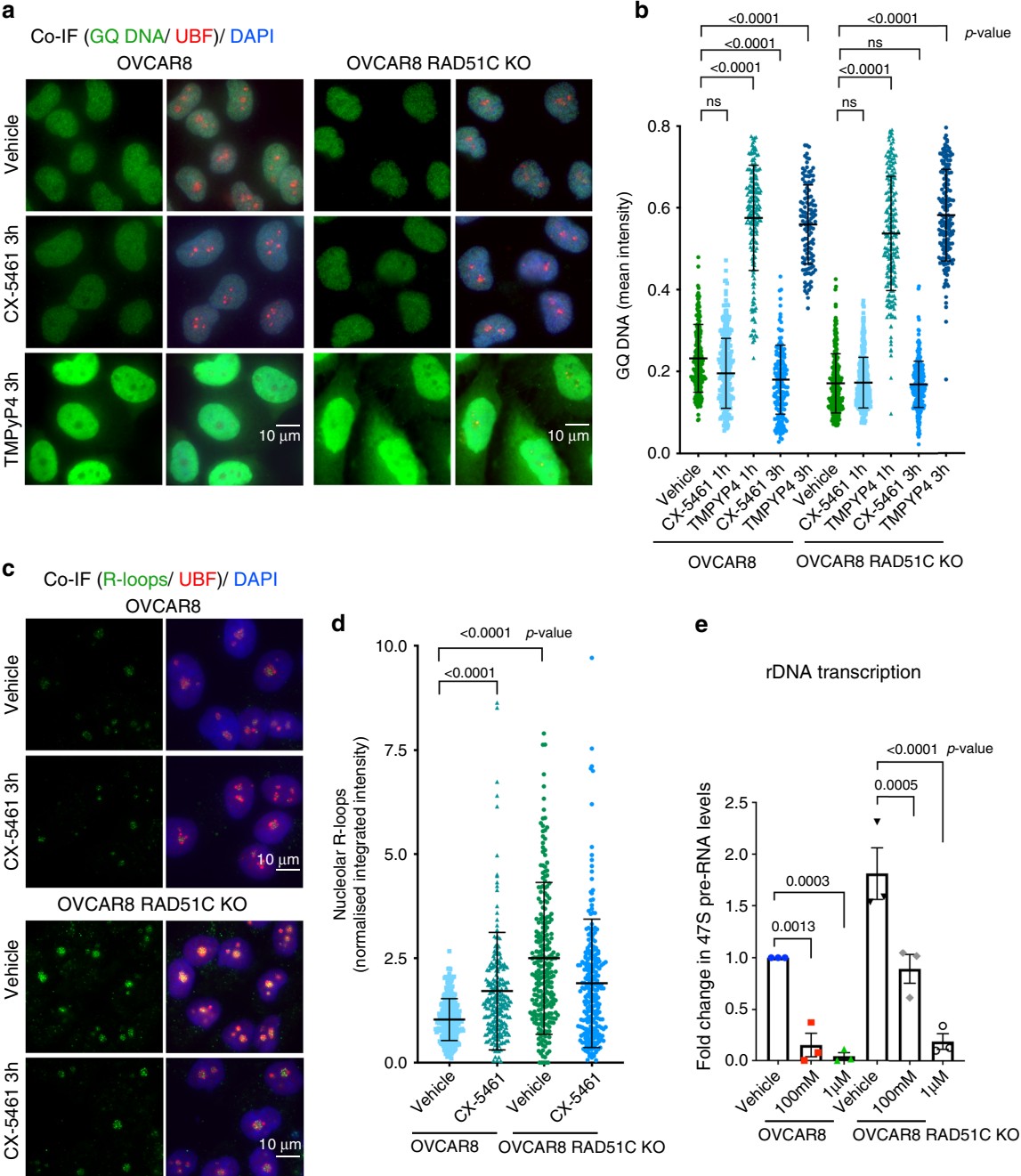

**Fig. 4 CX-5461 induces R-loops stabilization in HR-proficient OVCAR8 cells. a** Co-immunofluorescence (Co-IF) of GQ DNA and UBF as a nucleolar marker in OVCAR8 cells and OVCAR8 RAD51C KO cells treated with either vehicle, 1 μM CX-5461 or 10 μM TMPyP4 for 3 h. Representative images of three biologically independent experiments. **b** Quantitation of GQ DNA immunofluorescence. Signal intensities were analyzed using CellProfiler. Error bars represent mean ± SD, $n = 200$ cells per treatment condition examined over three independent experiments. **c** Co-IF analysis of R-loops and UBF in cells treated with vehicle or 1 μM CX-5461 for 3 h. Representative images of two biologically independent experiments. **d** Quantitation of R-loops signal intensity was performed using CellProfiler. $n = 230$ cells per treatment condition were examined over two independent experiments. Integrated intensity was normalized to corresponding median value of OVCAR8 vehicle control. Statistical analysis in (**b**) and (**d**) were performed using a one-sided one-way ANOVA, Kruskal–wallis multiple comparisons test (adjusted $p$-values are shown). (ns) denotes a non-significant $p$-value greater than 0.05. **e** Cells were treated with vehicle, 100 nM or 1 μM CX-5461 for 3 h. RNA was extracted and 47S rRNA precursor levels were determined using primers specific to the 5′ETS of pre-rRNA. Expression levels were normalized to NONO mRNA and expressed as a fold change relative to vehicle ($n = 3$ biologically independent experiments), error bars represent mean ± SEM, statistical analysis was performed using a two-sided one-way ANOVA, Tukey's multiple comparisons test (adjusted $p$-values are shown).

Furthermore, in contrast to CX-5461 treatment, TMPyP4 did not activate DDR at 1 h post-treatment at which time significant increase in GQ stabilisation was observed (Supplementary Fig. 4C-E). Our data clearly demonstrate that CX-5461-mediated activation of DDR is not associated with GQ stabilization. This finding contradicts a previous report in colon cancer cells[17] suggesting CX-5461 mediated effect on GQ stabilization may be cell type dependent.

We next examined the formation of RNA:DNA hybrids (R-loops), which are by-products of Pol I transcription. Stabilization of these three-stranded structures of nucleic acids consisting of a DNA-RNA hybrid and displaced ssDNA is known to obstruct DNA replication and activate DDR[32]. Recently, R-loops stabilization was demonstrated to coincide with Pol I transcription inhibition and activation of nucleolar DDR following mild hypo-osmotic stress[33]. To mark nucleolar R-loops we performed co-IF for the upstream binding transcription factor (UBF), which localizes to decondensed rDNA chromatin[34,35]. A striking change in nucleolar R-loop staining was observed, from being weak and diffuse in control OVCAR8 and OVCAR4 to a focal pattern with more intense foci in CX-5461 treated cells (Fig. 4c, d and Supplementary Fig. 5A-C). This suggests that CX-5461-induced Pol I displacement from rDNA promoters[19,36] is associated with stabilization of R-loops, possibly due to inhibition of initiation coinciding with inhibition of processing of precursor rRNAs leading to stabilization of R-loops from transcripts already generated by elongating Pol I molecules. Intriguingly, RAD51C KO OVCAR8 cells exhibited high basal level of co-transcriptional R-loops (Fig. 4c, d) coinciding with higher basal rate of rDNA transcription compared with parental cells (Fig. 4e). Thus, the data implicates RAD51C and/or HR activity in negative regulation of Pol l transcription. However, RAD51C depletion did not cooperate with 100 nM or 1 μM CX-5461 treatment in further reducing rDNA transcription or in enhancing accumulation of nucleolar R-loops compared with CX-5461-treated OVCAR8 cells.

To assess replication stress at rDNA, we examined whether CX-5461 induces ATR at the nucleoli (Fig. 5a). Indeed, pATR T1989 was detected following CX-5461 treatment at the periphery of nucleolar R-loops, co-localizing with UBF in CX-5461-treated HR-proficient OVCAR8 and OVCAR4 cells (Fig. 5a, b, Supplementary Fig. 5d-f and Supplementary Fig. 6a). Localization of ATR to the periphery of the nucleoli is consistent with the well characterised movement of damaged rDNA to anchoring points at the nucleolar periphery where it is more accessible to DNA repair factors[37]. Activation of ATR in response to CX-5461 was not restricted the nucleoli in S-phase cells and was induced in the EdU-negative and EdU-positive HR-defective cell populations (Fig. 5a, b). These data suggest that ATR is activated in response to CX-5461-induced defects in DNA replication, which occur in S-phase and in response to additional defects independent of the cell cycle. To examine whether activation of DDR in non-replicating cells is required for CX-5461-mediated growth inhibition, we analyzed the effect of CX-5461 treatment on sorted G1, S and G2 FUCCI-labelled OVCAR8 and RAD51C KO OVCAR8 cells (Supplementary Fig. 6B). Regardless of the cell cycle stage, CX-5461 treatment led to G2 cell cycle arrest. Due to the lack of p53 activity in OVCAR8 cells, the G2 checkpoint arrest appears to be the major response to CX-5461, suggesting DNA replication to be required for CX-5461 growth inhibitory effects.

RPA-coated ssDNA recruits and activates numerous DNA repair and cell cycle checkpoint regulators including ATR. ATR phosphorylates RPA32 at S33 immediately after fork stalling[31]. We therefore investigated S33 phosphorylation of RPA32 upon CX-5461 treatment and observed significant localization of pRPA32 S33 at the nucleoli in both HR-proficient and HR-deficient cells (Fig. 5c-d). CX-5461-mediated S33 phosphorylation of RPA was independent of the cell cycle stage and was not restricted to the nucleoli in HR-deficient cells. Thus, the formation of ssDNA structures in CX-5461 treated cells can lead to replication fork stalling and ATR activation with HRD exacerbating CX-5461-mediated replication stress and this may underpin CX-5461's synthetic lethal interaction with HRD.

We next assessed the contribution of R-loops stabilization to CX-5461-mediated toxicity by overexpressing ribonuclease H 1 (RNase H), which specifically degrades the RNA moiety in RNA: DNA hybrids[38] to prevent R loop stabilization in RAD51C KO OVCAR8 cells (Supplementary Fig. 7A). RNAse H overexpression reduced nucleolar R-loops levels in OVCAR8 RAD51C KO cells treated with CX-5461 compared with vehicle control, however this did not prevent CX-5461-mediated S33 phosphorylation of RPA within 3 h of treatment indicating the presence of ssDNA and fork stalling (Supplementary Fig. 7B, C). At 24 h post CX-5461 treatment, RNAse H overexpression partially reduced global replication stress marked by RPA32 S4/S8 phosphorylation, however this did not rescue CX-5461 growth inhibitory effects (Supplementary Fig. 7D, E). Altogether, our data implicates CX-5461 in inducing ssDNA structures and replication stress at rDNA, with stabilization of R-loops being indicative of chromatin defects at rDNA and contributing to CX-5461-mediated DDR but not being essential for CX-5461 efficacy.

**CX-5461 cooperates with PARPi in inducing replication stress.** We next examined γH2AX foci formation after 3-h treatment with CX-5461 at which time phosphorylation of ATR and RPA was detected. CX-5461 induced global γH2AX foci only in the EdU-positive population of HR-proficient OVCAR8 cells, suggesting that CX-5461-induced DNA damage is associated with DNA replication (Fig. 6a, b). HR-deficient RAD51C KO OVCAR8 cells treated with CX-5461 exhibited high levels of activation of ATR (Fig. 5b) and γH2AX foci (Fig. 6a, b) in both the EdU-positive and -negative populations. This data suggests that replication stress is not the only cause of CX-5461-induced DNA damage. However, our data utilizing sorted G1 cells using the FUCCI system (Supplementary Fig. 6b) demonstrated CX-5461-induced DDR had no effect on G1 cells, which progressed to G2 before arresting, suggesting DNA replication is required for CX-5461 growth inhibitory effects.

BRCA1/2 and RAD51 play major roles in replication fork stabilization following replication stress by preventing nucleolytic degradation of replication forks by the nuclease MRE11[39]. We therefore performed DNA fibre analysis to investigate the effect of CX-5461 on fork stabilization (Fig. 6c and Supplementary Fig. 8A) in OVCAR8 cells. Nascent replication tracks were sequentially labelled with CldU and IdU before treatment with CX-5461 for 3 h. CX-5461 treatment causes an overall decrease in track length, suggesting degradation of replication forks upon induction of DDR by CX-5461. This was rescued by co-treatment with the MRE11 inhibitor mirin, confirming inhibition of the MRE11 nuclease can rescue CX-5461-mediated fork destabiliza-tion. We next assessed whether DNA damage induced by CX-5461 treatment affects fork progression by pre-treating cells with CX-5461 for 24 h and then pulse labelled with both analogs (Fig. 6d). Pre-treatment with CX-5461 had no effect on fork length suggesting CX-5461 does not cause any lesions that could impact fork restarting or progression. On the other hand, the PARPi talazoparib (BMN-673) increased fork progression in agreement with a recent report implicating PARPi mediated acceleration of fork elongation as a mechanism for replication stress and DNA damage[40]. Thus, our data demonstrate that

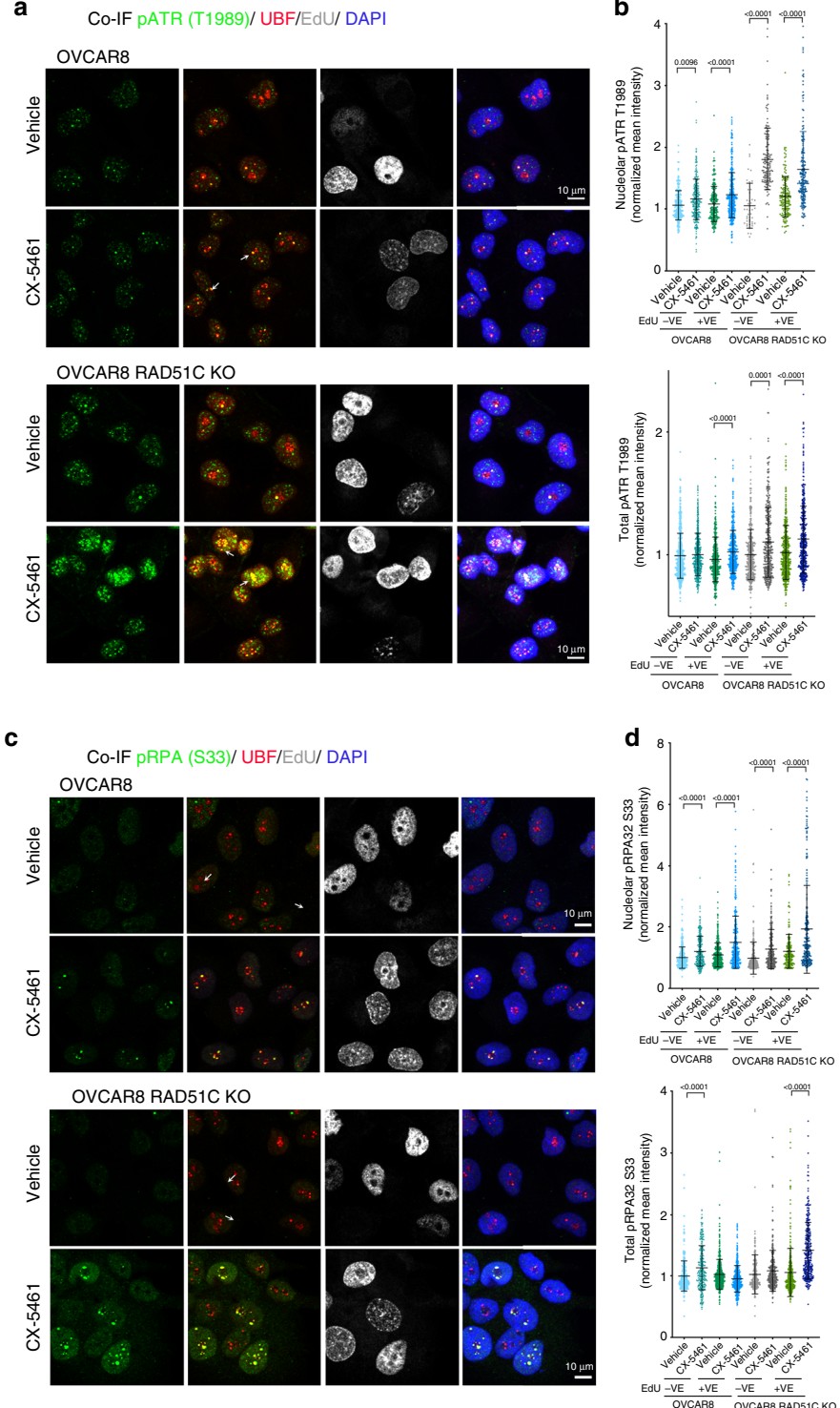

**Fig. 5 CX-5461 activates DDR. a** Co-IF analysis of pATR (T1989) and UBF in cells labelled with EdU and treated with vehicle or 1 μM CX-5461 for 3 h. Representative images of three biologically independent experiments. **b** Quantitation of signal intensity of pATR/UBF colocalized regions and total pATR was performed using CellProfiler and normalized to the median of vehicle treated controls. n = 464 EdU positive cells and n = 250 EdU negative cells per treatment condition examined over three biologically independent experiments. Error bars represent mean ± SD. Statistical analysis was performed using a two-sided one-way ANOVA, Kruskal–wallis multiple comparisons test (adjusted p-values are shown). **c** Co-IF analysis of pRPA32 (S33) and UBF in cells labelled with EdU and treated with vehicle or 1 μM CX-5461 for 3 h. Representative images of three biologically independent experiments. **d** Quantitation of signal intensity of pRPA/UBF colocalized regions and total pRPA was performed using CellProfiler and normalized to the median of vehicle treated controls. n = 216 EdU positive and n = 270 EdU negative cells per treatment condition examined over three independent experiments. Error bars represent mean ± SD. Statistical analysis was performed using a one-sided one-way ANOVA, Kruskal–wallis multiple comparisons test (adjusted p-values are shown).

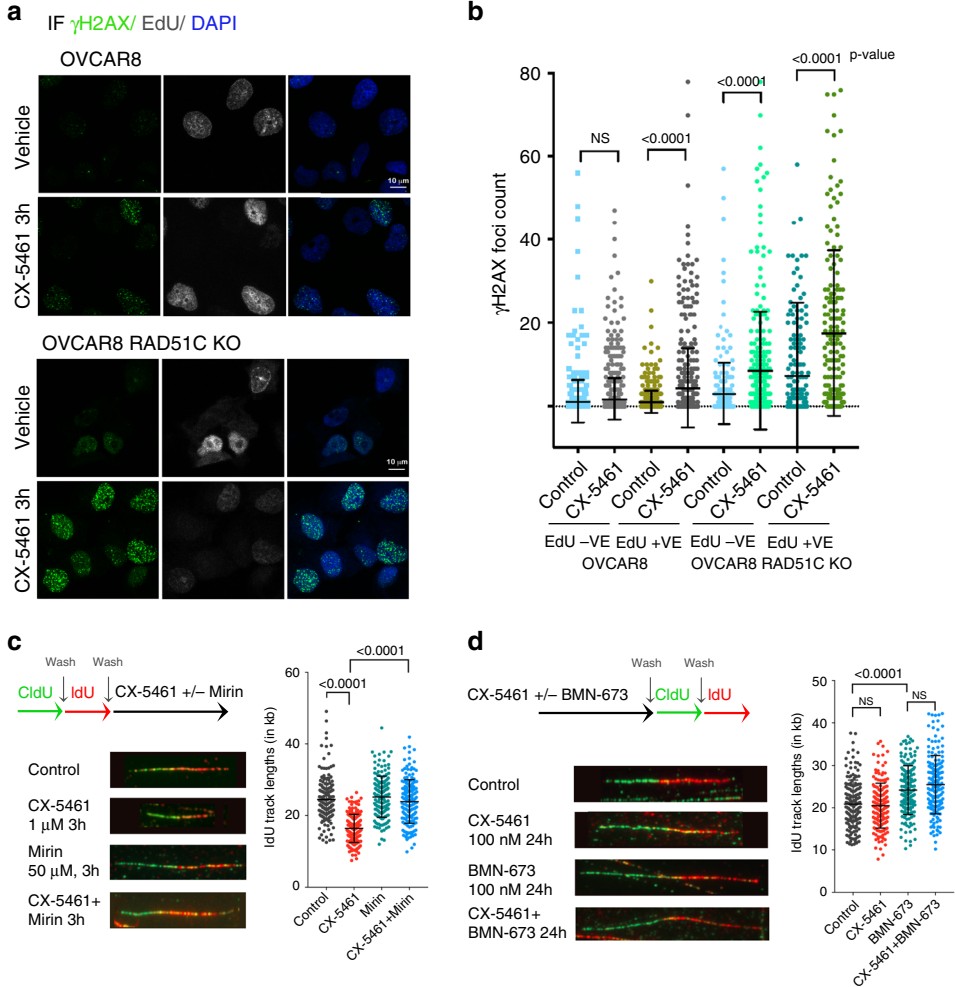

**Fig. 6 CX-5461-induces replication-dependent DNA damage in HR-proficient HGSOC cells. a** Co-IF analysis of γH2AX in cells labelled with EdU and treated with vehicle or 1 μM CX-5461 for 3 hours. Representative images of three biologically independent experiments. **b** Quantitation of foci count was performed using CellProfiler. n = 250 EdU positive cells and n = 220 EdU negative cells per treatment condition were examined over three independent experiments. Error bars represent mean ± SD. **c** IdU track length is reduced by CX-5461 through MRE11-dependent mechanism. Schematic of CIdU and IdU pulse-labelling method used (top). OVCAR8 cells were sequentially labelled and either processed or treated with 1 μM CX-5461, 50 mM mirin or the combination of both for 3 h. Fibres were processed for DNA fibre analysis. Replication Fork lengths were calculated based on the length of individual IdU tracks measured using ImageJ software. IdU track lengths (in μm) were converted to kb (1 kb = 2.59 μm). n = 150 replication tracks were analysed over two biologically independent experiments. Error bars represent mean ± SD. **d** DNA fibre analysis of OVCAR8 cells pre-treated with 100 nM CX-5461, 100 nM BMN-673 or in combination, washed then sequentially labelled with CldU and IdU as indicated in the schematic (top). Fibres were processed and analyzed as described above. n = 150 replication tracks were analysed over two independent experiments. Error bars represent mean ± SD. Statistical analysis (in **b**–**d**) was performed using a two-sided one-way ANOVA, Kruskal–wallis multiple comparisons test (adjusted p-values are shown). NS denotes a non-significant p-value > 0.05.

CX-5461 and PARPi cause replication stress via different effects on fork destabilization indicating independent synthetic lethal interactions with HRD. Moreover, the combination of CX-5461 and BMN-673 led to a significant increase in γH2AX foci formation in HR-proficient and HR-deficient cells (Fig. 7a, b) suggesting their cooperation in exacerbating replication stress and DNA damage. As the HR pathway is required to counteract replication stress by stabilizing stalled replication forks, we also investigated HR pathway activation following CX-5461 treatment. We found CX-5461 induced RAD51 foci formation, which indicated loading of RAD51 onto DSBs while the combination of CX-5461 and BMN-673 led to further enhanced RAD51 foci formation (Fig. 7a, b). Thus, CX-5461-mediated DDR activates HR, up to the stage of RAD51 loading and its cooperation with BMN-673 in exacerbating replication stress enhances HR pathway activation. Furthermore, consistent with CX-5461 activation

of MRE11-mediated fork degradation, CX-5461 induced pRPA S4/S8, a marker of stalled replication forks in HR-proficient but more robustly in HR deficient RAD51C KO OVCAR8 cells (Fig. 7C). In addition, CX-5461 cooperates with BMN-673 in inducing pRPA S4/S8 levels in HR-deficient OVCAR8 RAD51C KO cells. Thus, CX-5461, PARPi and HRD cooperate in enhancing replication stress. Taken together, our data are consistent with a model (Fig. 7D) by which CX-5461 inhibits Pol I recruitment leading to rDNA chromatin defects including formation of ssDNA and replication stress at rDNA. CX-5461 also induces global replication stress associated with stalling and destabilization of replication forks via MRE11 activity leading to DNA damage. HRD potentiates CX-5461-mediated replication stress. The combination of CX-5461 and PARPi further exacerbate replication stress and this underpins the PARPi/ CX-5461/ HRD synthetic lethal interactions (Fig. 7d).

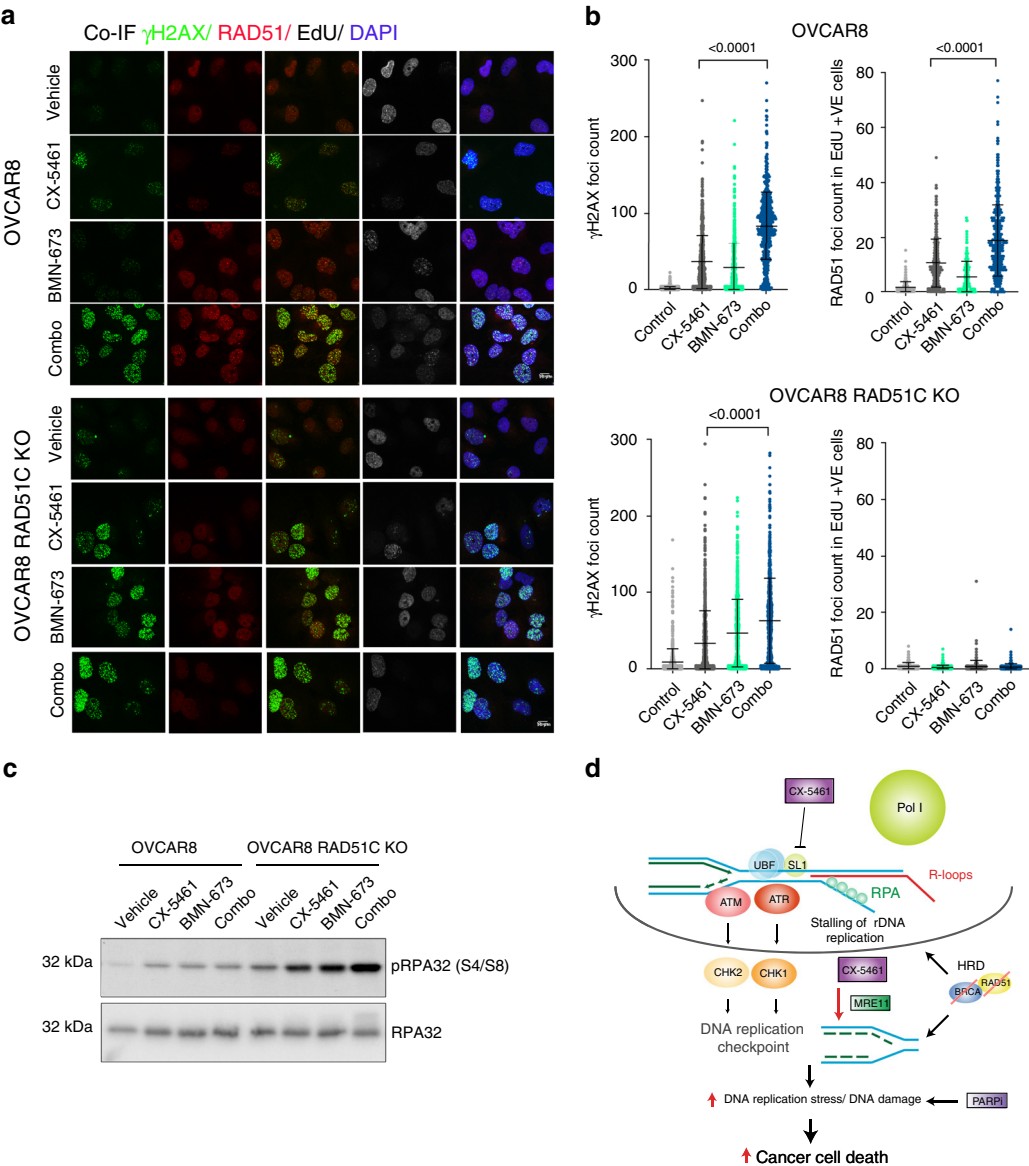

**Fig. 7 CX-5461 and BMN-673 induce markers of replication stress and DNA damage in OVCAR8 and OVCAR8 RAD51C KO cells. a** OVCAR8 cells were incubated with 10 μM EdU, before being treated with vehicle, 100 nM CX-5461, 100 nM BMN-673 or the combination of both for 24 h. Co-IF for γH2AX and RAD51 was performed. Cells were incubated for 30 minutes at room temperature with Click-IT reaction, washed with PBS and then counterstained with DAPI. Representative images of three biologically independent experiments. **b** Quantitation of γH2AX foci counts. $n = 554$ OVCAR8 cells and $n = 708$ OVCAR8 RAD51C KO cells per treatment condition were analysed over three biologically independent experiments. Error bars represent mean ± SD. Quantitation of RAD51 foci counts in EdU positive cells. $n = 223$ EdU +ve OVCAR8 cells and $n = 221$ OVCAR8 RAD51C KO cells per treatment condition analysed over three independent experiments. Error bars represent mean ± SD. Statistical analysis was performed using a two-sided one-way ANOVA, Tukey's multiple comparisons test (adjusted $p$-values are shown). **c** Western blot analysis of cells treated as in (**a**). Representative of $n = 2$ biologically independent experiments. The blots shown are of samples derived from the same experiment and were processed in parallel. Full scan sizes of western blots are provided in Supplementary Fig. 10. **d** A schematic of molecular response to CX-5461. CX-5461 inhibits the Pol I transcription complex by binding to the selectivity complex 1 (SL-1) and preventing Pol I from binding to rRNA gene promoters. Displacement of Pol I and inhibition of Pol I transcription initiation are associated with R-loops stabilization, recruitment of RPA to single strand rDNA, rDNA replication stress and activation of DDR at the nucleoli. CX-5461 also induces global replication stress associated with stalling and destabilization of replication forks via MRE11 activity leading to DNA damage, S-phase and G2/M cell cycle arrest. The HR pathway and PARP activity are necessary to counteract DNA replication stress. CX-5461 co-operates with HRD and inhibition of PARP activity in exacerbating replication stress and DNA damage, promoting cell death.

**CX-5461 cooperates with PARPi to inhibit HGSOC cell growth.** Since CX-5461 in combination with BMN-673 lead to increased DNA damage, we hypothesized that combining CX-5461 with PARPi or other DNA repair and DDR inhibitors (DDR therapy) may improve the efficacy of treating HGSOC. We therefore performed a focused/boutique drug screen in the HR-proficient OVCAR4 cells for DNA repair and DDR inhibitors

that may cooperate with CX-5461 to enhance growth arrest (Fig. 8a). Inhibitors of ATM (ATMi: KU55933), ATR (ATRi: VE-821), PARP (BMN-673), the platinum-based chemotherapy drug cisplatin, the mTORC1 inhibitor everolimus and the selective inhibitor of BCL-2 ABT-199 all demonstrated growth inhibitory effects as single agents. However, the combination of $GI_{20}$ dose of CX-5461 with BMN-673 and VE-821 (ATRi) showed the most

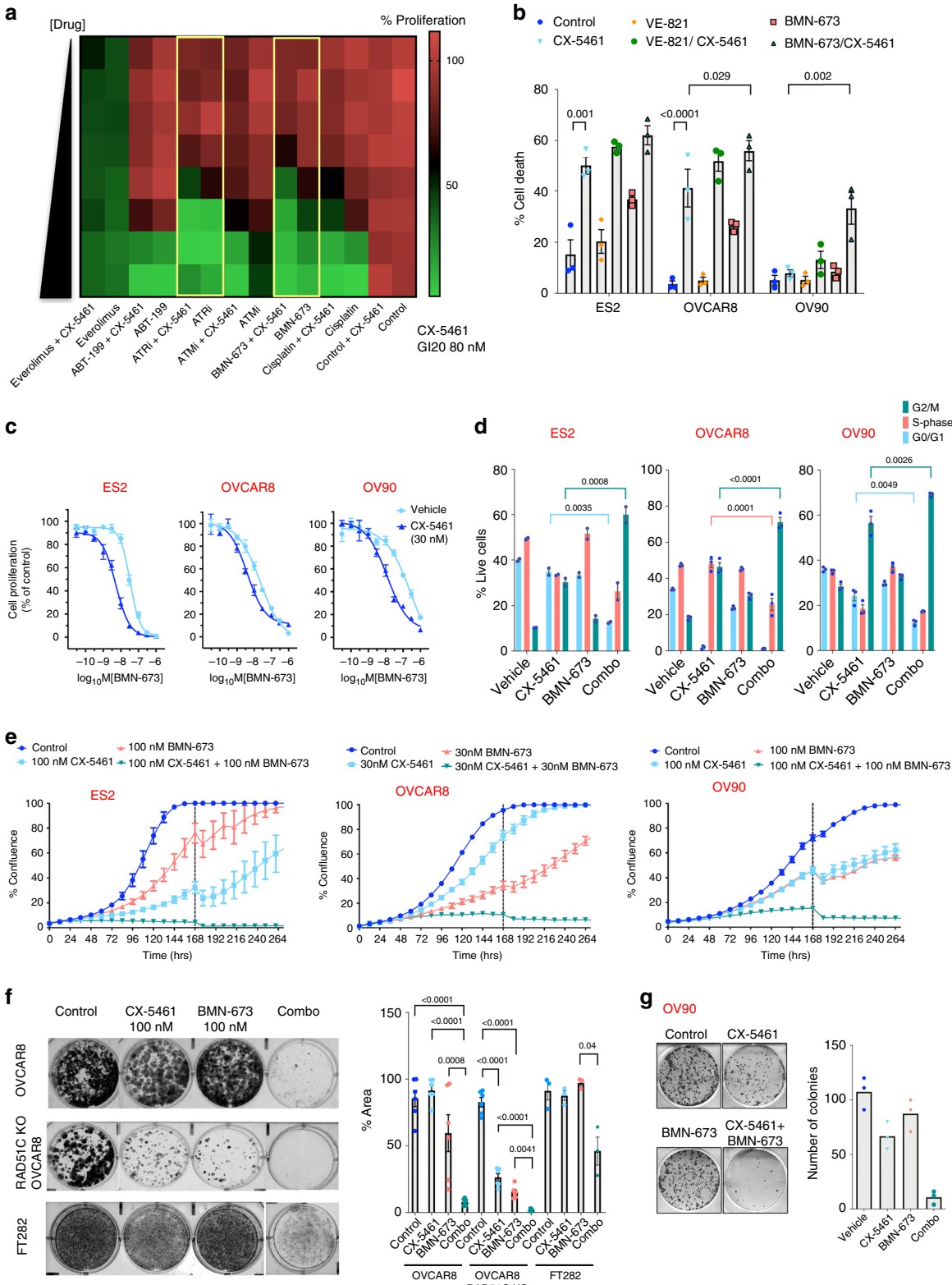

enhanced proliferative arrest compared with single-agent effects and compared with combinations with other compounds (Fig. 8a). We next examined the cooperation between CX-5461 and BMN-673 or ATRi in inducing cell death in three additional HR-proficient HGSOC cell lines (Supplementary Fig. 3C) and identified robust and significant interactions between BMN-673 and CX-5461 in inducing cell death compared with the ATRi (VE-821) (Fig. 8b). In addition, dose-response curves of BMN-

673 in the presence or absence of CX-5461 confirmed CX-5461's strong interaction with BMN-673 (Fig. 8c). Furthermore, the combination of CX-5461 and BMN-673 led to enhanced G2/M cell cycle arrest, enhanced inhibition of cell proliferation (Fig. 8d–e) and significantly reduced clonogenic survival of HR-proficient (OVCAR8 and OV90) and HR-deficient (OVCAR8 RAD51C KO) cells. In comparison, decreased sensitivity to the single agents and the combination in FT282 immortalised human

**Fig. 8 CX-5461 cooperates with PARPi in inhibiting HGSOC cell growth. a** Mini drug screen in OVCAR4 cells treated with increasing doses of cisplatin (0–1.11 μM), PARPi (BMN-673, 0–0.11 μM), ATMi (KU55933, 0–1.11 μM), ATRi (VE-821, 0–1.11 μM), ABT-119 (0–1.11 μM) or Everolimus (0–0.11 μM) ± 80 nM (GI$_{20}$) CX-5461. Colour-coding denotes the level of proliferation as measured by DAPI staining and imaging using Cellomics (green denotes reduced proliferation). Dose response of single drug treatments were corrected for vehicle control and the combination was corrected for response to 80 nM CX-5461, the average values of $n = 5$ are presented. The combination of CX-5461 with BMN-673 or ATRi is highlighted by yellow boxes. **b** Quantitation of SubG1 DNA content by PI staining of cells treated with vehicle, 100 nM CX-5461, 1 μM VE-821, 100 nM BMN-673 or in combination for 7 days ($n = 3$ biologically independent experiments). Error bars represent mean ± SEM. Flow cytometry gating strategy is shown in Supplementary Fig. 3D. C) Representative BMN-673 dose response curves ± 30 nM CX-5461. Cell proliferation was measured using SRB assays at 5 days post-treatment. Dose response curves for BMN-673 were corrected for DMSO treatment control and the combination was corrected for response to 30 nM CX-5461. Representative of three biological replicates. Error bars represent mean ± SEM of $n = 5$ technical replicates. **d** BrdU cell cycle analysis of cells treated with vehicle, 1 μM CX-5461, 100 nM BMN-673 or in combination for 72 h as described in Fig. 3b and Supplementary Fig. 3D. $n = 3$, mean ± SEM. **e** In vitro proliferation time-course assessed by cell confluency using IncuCyte ZOOM. Representative of $n = 3$ biological replicates, mean ± SEM of five technical replicates. Dashed lines denote re-supplement of media with drugs. **f** CX-5461 and BMN-673 cooperate in inhibiting clonogenic survival. Representative image of $n = 6$ biologically independent experiments for OVCAR8 and OVCAR8 RAD51C KO cells and $n = 3$ biological replicates for FT282 cells, mean ± SEM. **g** Clonogenic assay of OV90 cells. $n = 3$ technical replicates. Statistical analysis (in **b**, **d** and **f**) was performed using a two-sided one-way ANOVA, Tukey's multiple comparisons test (adjusted $p$-values are shown).

fallopian tube epithelial cells demonstrates a clear therapeutic window for these treatments (Fig. 8f, g).

Next, we investigated the effects of combining CX-5461 with BMN-673 on Pol I transcription rates based on the fact that nucleolar PARP1 is implicated in regulation of rDNA heterochromatin[41]. After 3-h treatment with CX-5461, we detected significant decreases in rRNA precursor levels (Supplementary Fig. 8B). However, the combination of CX-5461 with BMN-673 did not further decrease rRNA abundance compared with CX-5461-treated samples. Therefore, our data suggest CX-5461 cooperates with PARPi in inducing growth arrest and cell death by exacerbating replication stress and DNA damage (Fig. 7d) as opposed to enhancing inhibition of Pol I transcription.

**CX-5461 has significant therapeutic efficacy in HGSOC models.** We next investigated the potential of CX-5461 and PARPi interaction in vivo in a *BRCA2*-mutated, HR-deficient post one line of platinum treatment HGSOC-PDX (#19B). The administration of CX-5461 and olaparib as single agents resulted in stable disease and a statistically significant survival benefit (median time to harvest (at ethical endpoint) (TTH) for CX-5461 treatment 53 days, olaparib 67 days vs vehicle 22 days, $p$-values 0.00285 and 0.00285 compared with vehicle, respectively) (Fig. 9a). Co-treatment of CX-5461 and olaparib was well-tolerated (Supplementary Fig. 9A) and resulted in dramatic durable regression with reductions in tumour volumes indicating partial remission (defined as reduction in tumour volume of >30% from baseline) with survival lasting more than 100 days (median TTH 113 days, $p$-values 0.00692 compared with CX-5461 single agent treatment).

To compare the in vivo efficacy of CX-5461 with standard-of-care therapy, we examined the activity of CX-5461 in chemo-naïve HGSOC-PDX (#62) that exhibited *BRCA1* promoter hypermethylation and was previously characterized as resistant/refractory to cisplatin[44] and only responsive to a high dose (300 mg/kg) of the PARPi rucaparib. PDX#62 did not respond to 150 mg/kg rucaparib[29,42] and unlike PDX#19B, also did not respond to 50 mg/kg olaparib showing progressive disease with an increase in tumour volume of >20% from baseline at 8 days post-treatment (Fig. 9b). In comparison, CX-5461 therapy led to stable disease and a statistically significant survival benefit (median TTH for CX-5461 treatment 92 days vs vehicle 32 days and olaparib 36 days, $p$-values 0.0004 and 0.0022 compared with vehicle and olaparib, respectively). At a higher dose of olaparib (100 mg/kg) PDX#62 also showed progressive disease confirming its reduced response to olaparib (Supplementary Fig. 9B). In both

PDX#62 survival experiments, CX-5461 exhibited significant single agent therapeutic efficacy however the combination of CX-5461 and olaparib provided no additional survival benefit.

The previously characterized WEHICS62 cell line[29] generated from PDX#62 shows compromised HR capacity (Supplementary Fig. 3A and B). However, DNA fibre analysis showed, unlike effects observed in HR-deficient OVCAR8 RAD51C KO cells, forks stalled by hydroxyurea (HU) were not degraded in WEHICS62 cells, indicating fork protection (Fig. 9c, d), a mechanism associated with PARPi-resistance in BRCA1-deficient cancer cells[43]. CX-5461 treatment destabilized forks and the combination with HU enhanced fork instability. Thus, our data demonstrate CX-5461 acts to overcome fork protection in olaparib-resistant cells. CX-5461 treatment of WEHICS62 cells led to stabilization of nucleolar R-loops, indicative of rDNA chromatin defects (Supplementary Fig. 9C) however RNAse H overexpression only partially reduced CX-5461's induction of pRPA32 S4/S8 and did not rescue CX-5461 growth inhibitory effects (Supplementary Fig. 9D and E). Notably, CX-5461 induces activation of ATR, a robust G2/M cell cycle arrest and inhibition of proliferation (Fig. 9e–f and Supplementary Fig. 9F). Importantly, PDX#62 characterized as resistant/refractory to cisplatin, harbours amplifications in multiple cancer-associated genes and increased expression of Cyclin E and MYCN[42], known to be associated with resistance to platinum drugs. The therapeutic response to CX-5461 in PDX#62 is consistent with the MYC targets gene expression signature being associated with sensitivity to CX-5461 (Fig. 2a). Altogether, our data demonstrated CX-5461 has important clinical implications for the treatment of patients with olaparib-resistant OVCA and for patients with high MYC activity tumours and poor clinical outcome[1].

In order to assess the potential of CX-5461 therapy in HGSOC treatment, we next examined the prevalence of CX-5461 sensitivity signatures (Fig. 2a) in HGSOC tumours samples. We investigated 81 primary ovarian tumour samples (Fig. 10a) of which 21 (26%) exhibited the MYC_UP and BRCAm-mutated gene expression signatures of sensitivity to CX-5461. These stage III and IV tumours had various responses to chemotherapy: 2 progressed while receiving chemotherapy and 13 relapsed following therapy (Fig. 10a) suggesting, consistent with our pre-clinical data, that CX-5461 may have a different spectrum of sensitivity to chemotherapy in the clinic.

In addition, we investigated 25 recurrent (ascites) samples of which 4 samples (16%) exhibited the CX-5461 sensitivity signatures (Fig. 10b, c). Thus, CX-5461 has an exciting potential as a therapeutic option for a subset of relapsed OVCA. Intriguingly, the matching primary tumour of three of the

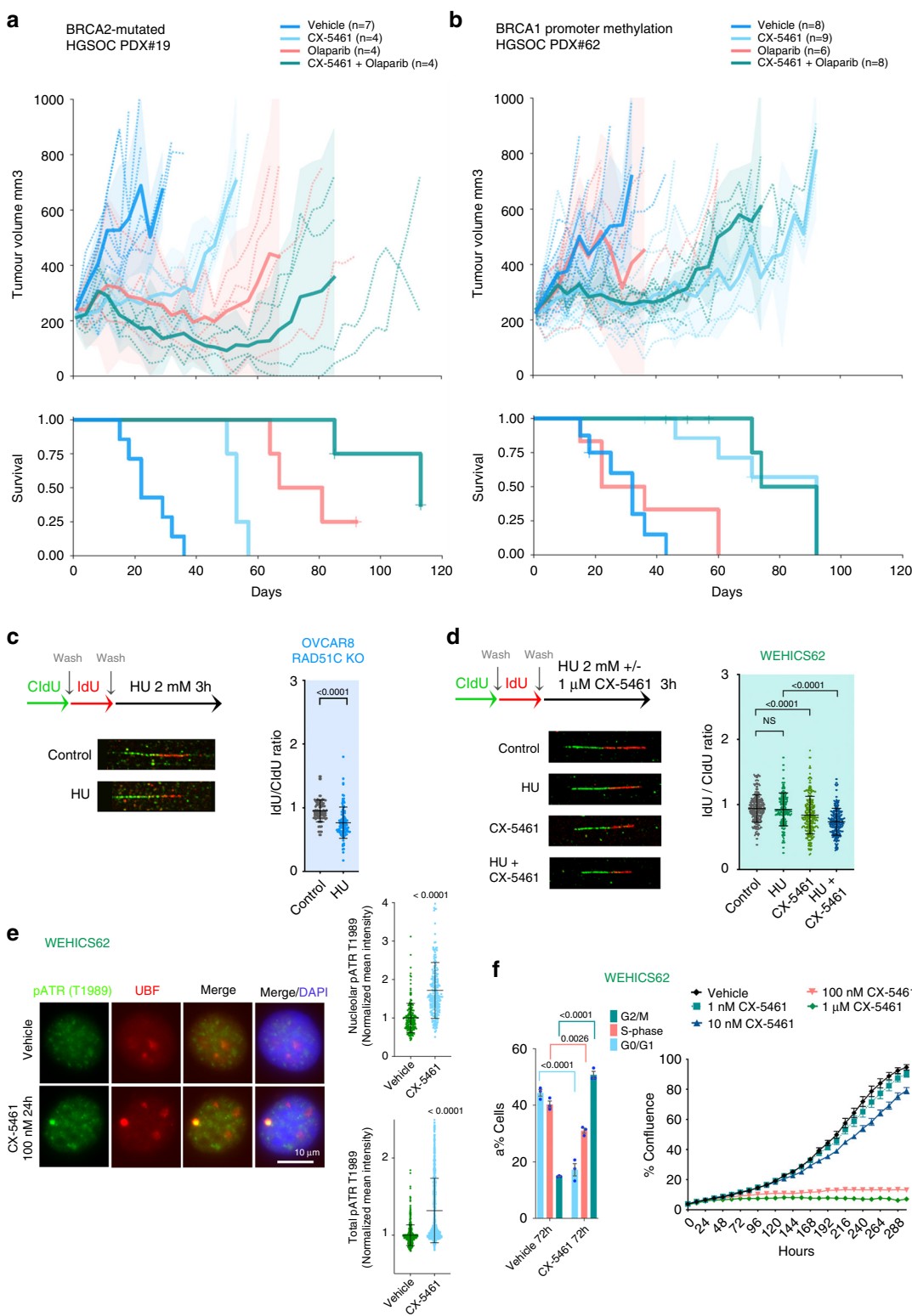

relapsed tumours predicted to respond to CX-5461 did not exhibit the CX-5461 sensitivity signatures (Fig. 10d, e). Thus, the development of resistance to chemotherapy is associated with changes that facilitate the efficacy of CX-5461.

## Discussion

We have previously demonstrated that CX-5461 effectively treats MYC-driven lymphoma[13,14] and AML independent of p53 status[15]. While induction of the p53-dependent impaired ribosome biogenesis checkpoint is a major mechanism of efficacy of CX-5461 in p53-wild-type tumours, activation of DDR is a key mechanism in the killing of p53-null AML and lymphoma[15,19].

In this report, we demonstrate that CX-5461 has single agent therapeutic efficacy against HR-deficient HGSOC. Importantly, we demonstrate that CX-5461 has significant therapeutic efficacy against a cisplatin- and olaparib-resistant HGSOC-PDX. We have identified BRCA-mutated and MYC targets gene expression signatures as biomarkers for sensitivity to CX-5461. In addition, we

**Fig. 9 CX-5461 has significant therapeutic efficacy in HGSOC- patient-derived xenografts (PDX) models. a** Responses observed in post-platinum treated BRCA2-mutant PDX#19 HGSOC-PDX and **b** PDX #62 with *BRCA1* promoter methylation to CX-5461 and olaparib treatment in vivo. Recipient mice bearing the PDX were randomized to treatment with vehicle, 40 mg/kg CX-5461 twice a week, 50 mg/kg olaparib once daily or CX-5461/olaparib combination for 3 weeks. The PDX were harvested at a tumour volume of 700 mm$^3$. Mean tumour volume (mm$^3$) (solid lines) ±95% CI (shaded region) and tumour volume of all individual mice (hashed lines) and corresponding Kaplan-Meier survival analysis. Censored events are represented by crosses on Kaplan-Meier plot. n indicates individual mice. **c** Schematic of CIdU and IdU pulse-labelling (top). OVCAR8 RAD51C KO cells or **d** WEHICS62 cell line derived from PDX#62[29] were sequentially labelled and either processed or treated with 2 mM hydroxyurea (HU) ± 1 μM CX-5461 for 3 h . Fibres were processed for DNA fibre analysis. n = 102 replication tracks of OVCAR8 RAD51C KO cells analyzed over two independent experiments, n = 236 replication tracks of WEHICS62 cells analysed over three independent experiments. Error bars represent mean ± SD. Statistical analysis in (C) was performed using a two-sided Mann–Whitney test and in (**d**) using two-sided one-way ANOVA, Tukey's multiple comparisons test (adjusted p-values are shown). NS denotes non-significant p-value. **e** Co-IF analysis of pATR (T1989) and UBF in WEHICS62 cells treated with vehicle or 100 nM CX-5461 for 24 h. Quantitation of signal intensity of the colocalized regions and total pATR was performed using CellProfiler. n = 506 cells per condition analysed over three independent experiments, error bars represent mean ± SD. Statistical analysis was performed using Mann–Whitney test. **f** BrdU cell cycle analysis of WEHCS62 cells treated with vehicle or 1 μM CX-5461 for 72 h (left panel), n = 3 biological replicates, mean ± SEM. Flow cytometry gating strategy is shown in Supplementary Fig. 3D. Statistical analysis was performed using a two-sided one-way ANOVA, Tukey's multiple comparisons test (adjusted p-values are shown). In vitro CX-5461 dose response proliferation time-course assessed using IncuCyte ZOOM. Representative of n = 3 biological replicates, mean ± SEM of five technical replicates.

have identified predictive signatures of CX-5461 sensitivity in 26% of primary and 16% of relapsed OVCA samples highlighting the potential of CX-5461 therapy in a subset of primary and acquired chemotherapy- and PARPi-resistant HGSOC. Specifically, we propose CX-5461 will have efficacy in HR-deficient HGSOC and in HGSOC tumours with elevated MYC activity such as the high-MYCN HGSOC subtype associated with poor prognosis[26,44].

Here, we also demonstrate that CX-5461 does not stabilize GQ structures in HGSOC cells, rather we show that by inhibiting Pol I transcription initiation, CX-5461 leads to recruitment of RPA to ssDNA and ATR activation at the nucleoli in HR-proficient cells. In HR-deficient cells, elevated nuclear pRPA and pATR and their recruitment to UBF-bound rDNA regions at the periphery of the nucleoli were observed independent of the cell cycle stage, indicating ATR activation by chromatin defects in addition to stalled replication forks at rDNA. Mechanistically, we demonstrate HRD potentiates CX-5461-mediated DDR identifying compromised HR-dependent resolution of global replication stress as the likely mechanism of CX-5461 synthetic lethal interaction with HRD in HGSOC. Furthermore, in agreement with our data, two recent reports found the sensitivity profile of CX-5461 to most closely resemble a TOP2 poison[21,22]. TOP2a is an essential component of the Pol I pre-initiation complex[23] and while our data clearly demonstrate CX-5461 inhibits Pol I transcription and activates nucleolar DDR, it is plausible that it does so by trapping TOP2 at rDNA and this perhaps influences TOP2 activity across the genome.

Our data also demonstrates CX-5461 causes stalling and destabilization of replication forks via MRE11 activity leading to replication stress, DNA damage and arrest of cell cycle progression. The net effect of CX-5461 destabilizing replication forks across the genome has important clinical implications. Recently, defects in stalled fork protection were identified as a common event (60%) in HGSOC patient-derived organoids[45]. Therefore, CX-5461 may have efficacy in a subset of HGSOC with functional defects in replication fork protection. Furthermore, our data demonstrate CX-5461-induced DDR overcomes fork protection in the HGSOC-PDX#62-derived cell line with reduced sensitivity to olaparib, a well-characterised mechanism of resistance to olaparib in BRCA-deficient cancer cells[11,43]. This highlights the potential of CX-5461 in the treatment of a subset of relapsed HGSOC.

The combination of CX-5461 and PARPi therapy showed robust therapeutic benefit in HR-deficient HGSOC, demonstrating that CX-5461's interaction with PARPi can significantly improve treatment of HR-deficient HGSOC. CX-5461 combination with PARPi led to increased replication stress, DNA damage and cell death, consistent with their distinct mode of action in destabilizing replication forks and inducing replication stress. In the absence of BRCA and RAD51, nascent replication forks are extensively degraded by MRE11. Thus, we propose that CX-5461 exacerbates HRD-mediated degradation of replication forks leading to increased replication stress and accumulation of DNA damage. Therefore, the combined effect of CX-5461, PARPi and HRD in enhancing replication stress through differential effects on replication fork stability leads to the accumulation of DNA damage that underpins their strong cooperation in promoting cancer cell death.

Altogether, our data provide evidence for the potential of combining CX-5461 and PARPi for improving the treatment of HR-deficient HGSOC. We demonstrate that CX-5461 enhances the synthetic lethal interaction of PARPi with HRD and clearly show that CX-5461 has a different mechanism of action to PARPi. Importantly, we characterized BRCA-mutated and MYC targets gene signatures as predictors of patient's response to CX-5461. MYC drives genome-wide transcription but among its main targets is Pol I transcription[25]. Indeed, we have shown MYC upregulation of Pol I transcription is required to drive malignant transformation in the Eμ-MYC lymphoma model[13,46]. Our data therefore suggest MYC-driven Pol I transcription and/or MYC-driven global transcription and replication stress underlie sensitivity to CX-5461. As CX-5461-sensitivity signatures were identified in primary and relapsed ovarian tumour samples, we propose that CX-5461 has exciting potential as a treatment option for patients with tumours harbouring HRD, unstable replication forks or high MYC activity who typically have poor clinical outcome and limited effective treatment options.

## Methods

**Compounds**. BMN-673, olaparib, KU55933, VE-821, ABT-199 and everolimus were purchased from Selleckchem. CX-5461 was purchased from Synkinase. For use in vitro, 10 mM stocks of CX-5461 were prepared in 50 mM NaH$_2$PO$_4$; 10 mM stocks of BMN-673, olaparib, KU55933, VE-821 and ABT-199 were prepared in DMSO. 1 mM stocks of everolimus were prepared in 100% ethanol.

**Cell lines**. The identity and individuality of ovarian cell lines used in this study (listed in Supplementary Table 1) were routinely confirmed by a polymerase chain reaction (PCR) based short tandem repeat (STR) analysis using six STR loci. The WEHICS62 cell line from PDX #62 and the OVCAR8 RAD51C KO cell lines[29] were provided by Prof Clare Scott (Walter and Eliza Hall Institute, Australia). Cell lines were cultured at 37 °C in 5% CO$_2$ in a humidified incubator and maintained in culture for a maximum of 8–10 weeks. Mycoplasma testing was routinely performed by PCR. OVCAR8, OVCAR4 and OVCAR8 RAD51C KO cell lines were

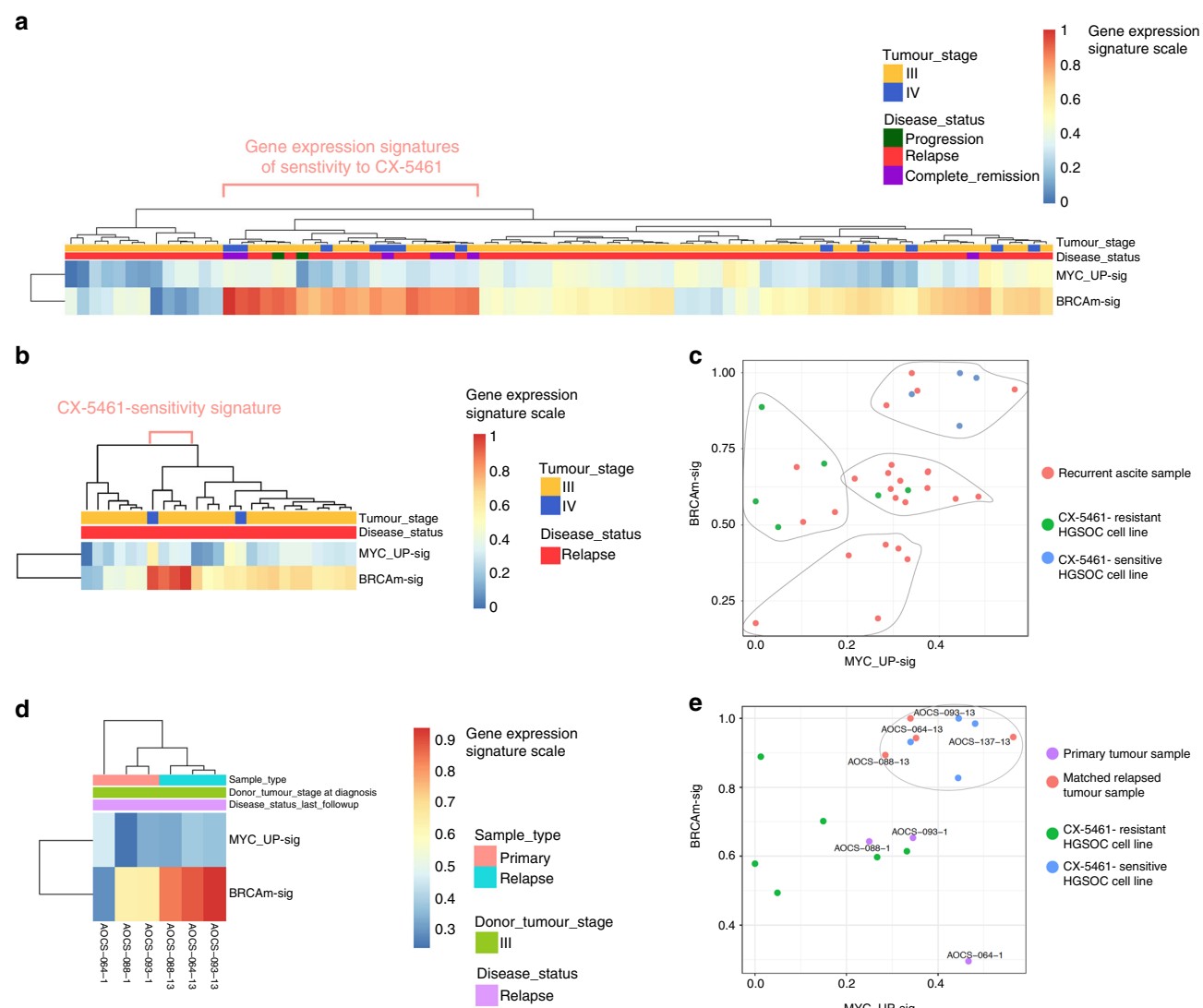

**Fig. 10 Detection of CX-5461-sensitivity gene expression signature in primary and relapsed HGSOC samples. a** Clinical and RNA-seq gene expression data from 81 primary ovarian tumour samples from the Australian cohort of ovarian cancer patients from the International Cancer Genome Consortium (ICGC) [https://dcc.icgc.org/] (release 27). The level of expression of the CX-5461 sensitivity signatures were calculated using ssGSEA in individual samples. ssGSEA scores were normalized by linear transformation to the 0–1 range for comparison. **b** ssGSEA scores were calculated as in (**a**) in 25 ascites samples from relapse patients from the ICGC. **c** Clustering of ascites samples from relapse patients with cell lines was based on the MYC_UP-sig and BRCAm-sig signatures using *k*-means with a *k* = 4. **d** Analysis of the MYC_UP-sig and BRCAm-sig in three matching primary tumour samples of three relapsed samples in (**c**) enriched with CX-5461-sensitivity signature. **e** Clustering of samples in E with cell lines samples based on the MYC_UP-sig and BRCAm-sig signatures using *k*-means with a *k* = 4.

grown in RPMI-1640 medium supplemented with HEPES, 10% (v/v) foetal bovine serum (FBS), 2 mM GlutaMAX (Gibco, 35050061)and 1% (v/v) antibiotics/anti-mycotics (Gibco, 15240062). The WEHICS62 cell line was grown in Dulbecco's modified Eagle's/ Ham's F-12 nutrient mixture (DMEM/F-12) containing 2 mM GlutaMAX, 5 µg/ml insulin, 50 ng/ml EGF, and 1 µg/ml hydrocortisone. FT282 cells were grown in DMEM/F-12 without HEPES in the presence of Ultroser G serum substitute (Biosepra, 15950-017). RNAse H overexpression construct[47] was kindly provided by Dr. Sonia Guil, Josep Carreras Leukaemia Research Institute, Spain). The Fluorescence Ubiquitin Cell Cycle Indicator (FUCCI)-labelled cell lines were produced by lentiviral transduction of pCSII-EF-mCherry-hCdt1(30/120) and pCSII-EF-mVenus-hGeminin(1/110) (kindly provided by Dr. Atsushi Miyawaki, RIKEN, Japan).

**Cell proliferation assays**. For assessing OVCA cells' sensitivity to CX-5461, dose response curves were generated for each cell line by plating cells in 96-well plates, culturing them for 24 hours and then treating them with either vehicle or increasing concentrations of drug (0, 0.001, 0.003, 0.01, 0.03, 0.1, 0.3, 1, 3, 10 and 30 µM CX-5461) for 48 h. Cell number was assessed using the IncuCyte ZOOM imaging system (Essen BioScience). For time-course assays examining cell

proliferation, cells were seeded (OVCAR8 at 500 cells; OVCAR8 RAD51C KO and WEHICS62 at 2000 cells per well) in 96-well plates, incubated overnight and then treated for up to 14 days, with drugs being replenished at 7 days. During this time the degree of confluency was measured every 24 h. To assess the anti-growth combination effects of CX-5461 and BMN-673, dose response curves were generated for the single agents using the sulforhodamine B (SRB) assay. In brief, cells were fixed with 50% (v/v) trichloroacetic acid (TCA) following drug treatment (5 days), stored for 1 h at 4 °C, washed with cold tap water then dried overnight. This was followed by adding 0.4% (w/v) SRB solution (Sigma-Aldrich, S1402) for 30 min at room temperature, repeatedly washing with 1% acetic acid then the remaining protein-bound dye was solubilized in 10 mM Tris base solution. The optical density measurements at 564 nm were quantitated using the iMark™ Microplate Absorbance Reader (BioRad). GI doses were determined using GraphPad Prism. For examining synthetic lethality of CX-5461 and siRNAs targeting the HR genes, individual siRNA duplexes (Dharmacon, GE lifesciences) were reversed transfected into OVCAR4 cells using Dharmafect 4 reagent (Dharmacon, GE lifesciences). 24 h later, transfection medium was changed to either CX-5461 (80 nM) or vehicle containing medium and cells were incubated further for 48 h. Cell proliferation was measured by cell count using DAPI staining and imaging using Cellomics CX7. Bliss Combination Index was calculated by dividing

combined viability (siRNA plus CX-5461 treatment) by the multiply of individual viability (siRNA only or CX-5461 only treatment). Bliss Combination Index lower than 0.9 is considered as synergy. siOTP (ON-TARGETplus Non-Targeting siRNA siRNAs) and siRNAs Dharmacon (catalogue No. siRAD54L (D-004592-17, D-004592-01, D-004592-02, D-004592-04); siRAD51AP1 (D-017166-01, D-017166-02, D-017166-03, D-017166-04); siBRCA2 (D-003462-04, D-003462-01, D-003462-02, D-003462-03).

**Cell cycle analysis**. For cell cycle analysis using 5-bromo-2′-deoxyuridine (BrdU) incorporation, cells were pulse labelled with 10 μM BrdU (Sigma-Aldrich, B5002) for 30 min then washed twice with PBS, collected, fixed in 80% ice-cold ethanol and stored at 4 °C until further processing. To perform staining, fixed cells were pelleted and incubated in 1 mL of 2 N HCl containing 0.5% (v/v) Triton X-100 for 30 min then pelleted and washed in 1 mL of 0.1 M Na$_2$B4O7.10H$_2$O (pH 8.5). Cell pellets were sequentially incubated for 30 minutes with anti-BrdU and FITC anti-mouse IgG antibodies (Supplementary Table 2) diluted in PBS containing 2% FBS and 0.5% Tween-20. Next, cells were washed with PBS containing 2% FBS and then incubated in 10 μg/mL propidium iodide (PI) solution at room temperature for 15 min. Lastly, cells were analyzed on the FACSCanto II (BD Biosciences) and cell cycle analysis was performed using Flowlogic software (Version 7.2.1, Inivai Technologies). Flow cytometry analysis of FUCCI-labelled cell lines' fluorescent protein signals was performed on FACSCanto II (BD Biosciences) and cell cycle analysis was also performed using Flowlogic software.

**Cell death assay**. Cell death was determined using PI staining followed by flow cytometry (FACSCanto II) and data analyzed using Flowlogic software.

**Clonogenic assays**. Cells were seeded in 6-well plates for 24 h. Following drug incubation for 5 days, media was aspirated, cells were washed and then incubated with drug-free media for 7 days. Next, cells were fixed with 100% methanol for 1 h, stained with 0.1% (w/v) crystal violet (Sigma-Aldrich, C6158) for 1 h, washed with H$_2$O and air dried. Colonies were counted manually using a stereo microscope or imaged and quantitated using the Chemidoc Imaging system (BioRad).

**Western blotting (WB)**. Twenty to fifty micrograms of whole-cell lysates were resolved by SDS-PAGE, electrophoretically transferred onto Immobilon-P poly-vinylidene fluoride (PVDF) membranes (MERCK Millipore, IPVH00010) and analyzed using enhanced chemiluminescence (ECL) detection (GE Healthcare, RPN2106). Antibodies details are listed in Supplementary Table 2. Full sized scans of western blots are provided in Supplementary Fig. 10.

**Gene expression analysis**. For Reverse-transcription qPCR analysis, cells were lysed, RNA extracted, and first-strand cDNA synthesized using random hexamer primers (Promega, C1181) and Superscript III (Invitrogen,18080044). Quantitative PCR (qPCR) was performed in duplicate using the FAST SYBR Green dye (Applied Biosystems, 4385610) on the StepOnePlus real-time PCR system (Applied Biosystems). Primer sequences are listed in (Supplementary Table 3).

Publicly available baseline gene expression microarray data of OVCA cell lines (GSE43765) provided by our group was utilised in this study. OVCA cells were harvested at 50–80% confluency (three biological replicates). RNA was extracted, in vitro transcribed and biotin labelled cRNA was fragmented and hybridized to Affymetrix 1.0ST expression array as per manufacturer's instructions. Differential gene expression was determined using the Limma R package (version 3.3.2) after RMA normalisation and back-ground correction[48]. Genes that had a >1.4-fold change in expression between resistant and sensitive were included in GSEA 1000 analysis, performed using the default weighted enrichment statistic and a signal-to-noise metric to rank genes based on their differential expression across sensitive and resistant cell lines[49].

We used ssGSEA[50] from the GSVA[51] package (version 1.20.0) in R (version 3.3.2) to obtain the level of activity of the HRD gene signature[24] (Supplementary Data 1) in individual samples. Here, genes in each sample were ranked according to their expression levels, and a score for each pathway was generated based on the empirical cumulative distribution function, reflecting how highly or lowly genes of a pathway are found in the ranked list. Statistical significance of the ssGSEA scores of different cell line categories (sensitive or resistant) was obtained using two-sided Wilcoxon tests.

**Immunofluorescence**. For IF assays combined with EdU labelling, cells were first incubated with 10 μM EdU for 30 min prior to drug treatment. Cells were fixed in 4% paraformaldehyde (PFA) (10 min at room temperature), permeabilized with ice cold 0.3% Triton X-100 in PBS for 10 min, washed with PBS, and then blocked with 5% goat serum and 0.3% Triton X-100 in PBS for 30 minutes at room temperature. Next, cells were sequentially incubated with primary antibody and secondary antibodies (Supplementary Table 2) at 37 °C for 1 h in a humidified chamber. Lastly, cells were incubated for 30 minutes at room temperature in Click-IT reaction (100 mM Tris pH 8.5, 10 nM Alexa Fluor 647-azide (Invitrogen, A10277), 1 mM CuSO4, and 100 mM ascorbic acid), then washed with PBS. Nuclei were counterstained with Vectashield mounting media containing DAPI.

For IF detection of GQ DNA using the 1H6 antibody, cells were treated with 40 μg/ml RNase A (Thermo Scientific, EN0531) for 1 h prior to the blocking step. For IF using the S9.6 (R-loops) antibody, cells post-PFA fixation were permeabilized with 100% methanol for 10 minutes and 100% acetone for 1 min on ice, washed with PBS prior to the blocking step. Images were acquired on an Olympus BX-61 microscope equipped with a Spot RT camera (model 25.4), using the UPlanAPO 60×, NA 1.2 water immersion objective and the Spot Advanced software (version 5.6). Settings for adjusting the image after acquisition (i.e. gamma adjust and background subtract settings) were identical for all images. Confocal images were acquired using Zeiss Elyra 63X magnification. Images were analyzed using CellProfiler version 3.1.9 (Broad Institute).

**DNA fibre analysis**. Exponentially growing cells were pulse-labelled for 20 min with 50 μM CldU (Sigma-Aldrich, C6891), washed three times with warm PBS and then incubated with 250 μM IdU (Sigma-Aldrich, 17125) for 20 min. After exposure to the second nucleotide analog, cells were washed again in warm PBS and either processed or treated for 3 h with 1 μM CX-5461, 50 μM Mirin (Sigma-Aldrich, M9948) or the combination of both. Labelled cells were trypsinized and resuspended in ice-cold PBS at $7.5 \times 10^5$ cells/mL. Two microlitres of this suspension were spotted onto a pre-cleaned glass slide and lysed with 10 μL of spreading buffer (0.5% SDS in 200 mM Tris-HCl, pH 7.4 and 50 mM EDTA) in a humidified chamber. After 36 min, the slides were tilted at 15° relative to horizontal, allowing the DNA to spread down the slide. Slides were air-dried, fixed in methanol and acetic acid (3:1) for 10 min and air-dried. DNA was denatured with 2.5 M HCl for 60 min at room temperature. Slides were then rinsed in PBS twice and blocked in PBS containing 0.1% Triton X-100 (PBS-T) and 1% BSA for 1 h at room temperature. Rat anti-BrdU (1:200, Abcam, ab6323) was applied overnight at 4 °C in a humidified chamber. Slides were then washed with PBS and incubated with Alexa Fluor 488-conjugated chicken anti-rat antibody at 1:200 dilutions (Invitrogen, A21470). Slides were washed with PBS and incubated for 45 min at room temperature with mouse anti-BrdU (Becton Dickinson, 347580) antibody at 1:50 dilution to detect IdU tracks. Slides were washed in PBS and stained with Alexa Fluor 594-labelled goat anti-mouse antibody (Life technologies, A-11030) at 1:300 dilutions at room temperature for 30 minutes. Slides were washed in PBS and mounted in Prolong Diamond antifade (Invitrogen, P36961). Replication tracks were imaged on a Deltavision microscope at 60× and measured using ImageJ software (1.47v, NIH). For studies with CX-5461 and BMN-673, the cells were treated for 24 h with individual drugs or in combination before cells were labelled with the BrdU analogs. Fibre assays in Fig. 9d were performed using silanized coverslips (Genomic Vision, COV-002) and the Molecular Combing System (MCS-001) from Genomic Vision (described in Supplementary methods).

**Animal studies**. All experiments involving animals were approved by the Walter and Eliza Hall Institute of Medical Research Animal Ethics Committee. PDX were generated from patients with OVCA enroled in the Australian Ovarian Cancer Study. Informed consent was obtained from all patients, and all experiments were performed according to the human ethics guidelines. Ethics approval was obtained from the Human Research Ethics Committees at the Royal Women's Hospital and the Walter and Eliza Hall Institute.

The housing facility was kept at 21 °C, with a relative humidity of around 50%. The light/dark cycle was 14 h light/10 h dark. PDX were generated as published previously by transplanting fresh fragments subcutaneously into NOD/SCID/IL2Rγnull recipient mice (T1, passage 1)[42]. Briefly, for PDX #19B fresh tumour fragments were subcutaneously implanted under the right flank. For PDX #62 frozen tumour fragments from previously passaged PDX[42], stored in DMSO supplemented media, were thawed and subcutaneously implanted under the right flank. Recipient mice bearing T4-T7 (passage 4 to passage 7) tumours were randomly assigned to treatment with olaparib, CX-5461, combination or vehicle when tumour volume reached 180–300 mm$^3$. Olaparib was administered once daily intraperitoneally at a dose of 50 mg/kg in vehicle (phosphate buffered saline (PBS) containing 10% DMSO and 10% 2-hydroxy-propyl-β-cyclodextrin). CX-5461 was given by oral gavage twice a week for 3 weeks at 40 mg/kg in vehicle (25 mM NaH$_2$PO$_4$ pH7.4). Tumours were measured twice per week and recorded in StudyLog software (StudyLog Systems). Mice were euthanized once tumour volume reached 700 mm$^3$ or when mice reached ethical endpoint. Nadir (taken as the smallest average tumour volume recorded since treatment started or 0.2 cm$^3$ if nadir was <0.2 cm$^3$), time to harvest (TTH) and treatment responses are as defined in ref. [42]. Median TTH was calculated by including censored events for PDX where mice were harvested when tumour volume was >500 mm$^3$ but <600 mm$^3$. Partial response was achieved if the average tumour volume reduced to between 50 and 140 mm$^3$ (>30% reduction from nadir, assigned as 200 mm$^3$) for two or more consecutive measurements. Tumour volume and survival graphs were produced with SurvivalVolume v1.2[52].

**Analysis of ovarian tumour samples**. We calculated the level of expression of the MYC_UP (MYC oncogenic Signature UP) and BRCAm (BRCA1 mutated UP) signatures (Supplementary Data 2&3) using ssGSEA in RNA expression data from the Broad Institute Cancer Cell Line Encyclopedia (CCLE), version 20180502 [https://portals.broadinstitute.org/ccle/data]. In cases of multiple entries for the

same gene ID, the median expression was used. To determine whether the inclusion of the MYC target genes signature provided additional power to predict CX-5461 sensitivity compared with BRCAm signature, we built two generalised linear models, one with the BRCAm signature and the other encompassing both signatures. We performed an ANOVA test to assess which model better predicted sensitivity in the CCLE data set of gene expression and CX-5461 drug sensitivity. The ssGSEA scores of the signatures in individual samples were then calculated from the RNAseq gene expression data of 81 primary solid tumour samples and 25 ascites samples from relapse patients from the Australian cohort of OVCA patients available from the International Cancer Genome Consortium [https://dcc.icgc.org/] (release 27). Only coding genes were considered and scores were normalized by linear transformation to the 0–1 range for comparison across data sets.

Clustering of relapse samples with cell lines was based on the BRCAm and MYC_UP signatures of samples using $k$-means with a $k = 4$.

**Reporting summary.** Further information on research design is available in the Nature Research Reporting Summary linked to this article.

## Code availability

The code and additional information used to generate Figs. 2a–c, 10 and Supplementary Fig. 2, including the CX-5461-sensitivity signatures, are available at [https://github.com/esanij/CX-5461-sensitivity-signature-in-ovarian-cancer].

## Data availability

Baseline gene expression microarray data of OVCA cell lines is available in a public repository (accession number GSE43765) from [https://www.ncbi.nlm.nih.gov/geo/query/acc.cgi?acc=GSE43765]. Gene expression data are publicly available from the original sources, CCLE [https://data.broadinstitute.org/ccle/CCLE_DepMap_18Q2_RNAseq_RPKM_20180502.gct] and from ICGC via the link [https://dcc.icgc.org/search?filters=%7B%22donor%22:%7B%22availableDataTypes%22:%7B%22is%22:%5B%22exp_seq%22%5D%7D,%22projectId%22:%7B%22is%22:%5B%22OV-AU%22%5D%7D%7D%7D] and selecting "Download Donor Data" followed by "Sequencing-based Gene Expression". IC50s from the Genomics of Drug Sensitivity database used for Supplementary Fig. 2 are also publicly available from [https://www.cancerrxgene.org, version v17.3]. The source data underlying Figs. 2a–c, 10 and Supplementary Fig. 2 are available at [https://github.com/esanij/CX-5461-sensitivity-signature-in-ovarian-cancer]. All other data supporting the findings of this study are available within the article and its supplementary information files and from the corresponding author upon reasonable request. A reporting summary for this article is available as a Supplementary Information File. Datasets generated and/or analysed during the current study are available from the corresponding author.

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

## Acknowledgements

We thank Jason Ellul for assistance in analysing OVCA cell lines gene expression data. We thank the Peter MacCallum Cancer Centre FACS Facility, Victorian Centre for Functional Genomics, Centre for Advanced Histology and Microscopy, Laboratory Services and Media Kitchen. This work was supported by the National Health and Medical Research Council (NHMRC) of Australia project grants and a NHMRC Program Grant (#1053792 and #1162052). Researchers were funded by NHMRC Fellowships (K.K., G.A.M., R.D.H., R.B.P.). This research was also supported by the Australian Cancer Research Foundation for the Peter Mac Centre for Advanced Histology and Microscopy facility.

## Author contributions

Conception and design: E.S., K.H., K.E.S. and R.B.P. Development of methodology: .S., K.H., S.Y., J.X., K.T.C., J.K., S.E., C.C., M.W. and E.B. Acquisition of data: E.S., K.H., J.X., N.B., J.A., J.S., O.K., E.L., P.N., D.F. Analysis and interpretation of data (e.g., statistical analysis, computational analysis): E.S., K.H., A.T., J.K., K.K., G.P., L.M., A.D., G.McA., J.S., R.D.H., C.S., K.S. and R.B.P. Writing, review, and/or revision of the manuscript: E.S., K.H., C.C., G.P., K.K., R.D.H., C.S.K. and S.R.B.P. Study supervision: E.S., K.H., K.S. and R.B.P.

## Competing interests

J.S, is Chief Medical Officer at Shenwa Biosciences Inc. R.D.H. is a Chief Scientific Advisor to Pimera Inc. G.A.McA. has commercial research grants from Celgene and Pfizer. No potential conflicts of interest were disclosed by the other authors.
