## [Peer Review File · Nature Communications]

Reviewers' comments:

Reviewer #1 (Remarks to the Author); expert in DNA damage:

In this manuscript Sanji et al. explore the action of the RNA polymerase I inhibitor CX-5461 in high-grade serous ovarian cancer (HGSOC) and its relationship with deficiencies in homologous recombination (HR) and resistance to treatment with PARP inhibitors. Although not entirely novel, as it builds on previous work from the laboratory in other cellular and in vivo cancer models, the extension to HGSOC is interesting. Furthermore, they provide some advance in the understanding of the mechanistic details responsible for the sensitivity. In general terms, the cancer therapy-related aspects of the manuscript are very solid, but the authors are too ambitious in their explanation of the molecular bases of CX-5461 action, which seem too complex to be satisfactorily explained with the results presented. Thus, some of the claims would need to be further substantiated, or alternatively, presented in a more descriptive fashion.

General comments:

1. Regarding the observed accumulation of R-loops. It is not easy for me to understand how an inhibition of transcription initiation, as caused by CX-5461, can generate an increase in R-loops. Authors propose that the absence of Pol I at the promoter favors R-loop formation, but in most models the formation of the R-loop is co-transcriptional, and would therefore be reduced upon transcription inhibition. As a matter of fact, in HR-deficient cell lines, CX-5461 causes a reduction R-loops (Figure 3D), which the authors fail to discuss appropriately. Furthermore, there is no correlation between the levels of R-loops and replication stress markers cell death. Therefore, the involvement of R-loop accumulation in CX-5461 toxicity, and the connection with replication stress that authors claim does not seem sufficiently substantiated. This could be done, for example, by analyzing a possible suppression of the replication stress and cell death-phenotypes upon RNase H overexpression.

2. The activation of ATR by CX-5461 treatment in non-replicating cells is not explained by a model that is dependent on replication stress and fork stability. This should be, at least, discussed in more detail. In addition, it could be useful to determine whether DNA replication is required for CX-5461 toxicity. If this were the case, the effects observed on non-replicating cells would be irrelevant from a therapeutic perspective and could be obviated.

3. Regarding the DNA fiber analysis of DNA replication. This technique measures global DNA replication, and not specifically at the rDNA. The effects observed are therefore unlikely to be directly caused by problems locally occurring at the rDNA as a consequence of inhibiting transcription by CX-5461. Authors need to address this important limitation; otherwise, the conclusions drawn from the analysis of replication are seriously compromised. In addition, an analysis of replication-fork asymmetry would be more appropriate to unambiguously claim an effect on replication-fork stability. Finally, I fail to see how the results presented demonstrate an effect of CX-5461 in overcoming fork protection in olaparib-resistant cell lines. This should be directly addressed by, for example, checking the effect of CX-5461 on fork stability in HU-treated WEHICS62 cells.

Specific comments:

4. For the analysis of CX-5461 sensitivity in HR-depleted cells (Figure 1H), the absolute numbers, and not just the combination index should be presented.

5. The claimed G2 accumulation in response to CX-5461 treatment (Figure 2B) is difficult to perceive in the way the data are presented.

6. The quality of the Western blot presented in Figure 2C is not sufficient. In addition, the effects claimed are not easily observed. For example, the accumulation of pCHK1/2 and gH2AX.

7. The accumulation of gH2AX and RAD51 foci in response to CX-5461, BMN-673 and their combination should also be analyzed in HR-deficient cells.

Reviewer #2 (Remarks to the Author); expert in mouse models and PARPi resistance:

This is an interesting study evaluating the impact of the RNA polymerase I inhibitor CX-5461 in high grade serous ovarian cancer (HGSOC). They demonstrate that this drug has more profound impact on DNA damage responses and replication fork degradation in HGSOC tumors with homologous recombination defects (HRD) and further show combinatorial effects with different DNA damage inhibitors including an ATR inhibitor, and ATM inhibitor, and a PARP inhibitor. They provide nice in vivo data demonstrating combinatorial activity for CX-5461 and a PARP inhibitor in a BRCA-deficient tumor model, and they provide a nice analysis of patient gene expression data to suggest that a reasonable fraction of patients would be anticipated to benefit from CX-5461 therapy. Overall, the studies appear carefully performed and are clearly presented. The results have broad implications for understanding how DDR inhibitors could be combined, how PARP inhibitors affect DDR, and the essential role for replication for instability in sensitivity to these inhibitors.

The key weakness in this paper that makes the manuscript difficult for this reviewer to easily understand is a failure to connect rDNA replication stress to a more global replication stress signal. Given the relatively small representation of rDNA to the overall size of the genome, it is not clear if the authors are arguing that this small region of replication stress is causing the over-arching effects on DDR signaling, cell death and DNA synthesis (CldU/IdU assays) of the RNA PolI inhibitor. Many of these assays will assess both ribosomal and non-ribosomal DNA regions. Thus, are there other mechanisms beyond effects on rRNA synthesis and rDNA replication stress that contribute to the cytotoxicity effects of the drug or are there off-target effects of the drug that also influence the efficacy.

The only significant issue identified in the data presentation and discussion is Figure 5A EdU staining of CX-5461 treatment – the EdU staining that outlines two nuclei does not seem to match the DAPI stained nuclei, suggesting this image is not of the same microscopic field as the other two images shown (DAPI and H2AX) for this sub-figure. This should be addressed and/or corrected.

Reviewer #3 (Remarks to the Author); expert in RNA polymerases:

Sanij and colleagues have presented a comprehensive and interesting paper investigating the RNA polymerase I inhibitor, CX-5461, including how it enhances the effect of PARP inhibitors. It is stated (earlier publications) that CX-5461 has shown efficacy in both p53 wild-type and mutant malignancies, functioning as a p53 independent activator of the DNA damage response. The data presented in this paper supports this as there was no statistical difference between wt or mutant p53 cell lines in response to CX-5461.

RNA pol I inhibition has also been previously published to show synthetic lethality with BRCA mutant cells. The in vivo work presented, including that CX-5461 and the PARPi olaparib given together was well tolerated and had additional effect over olaparib alone is promising. Further, that CX-5461 may have benefit when given as a single agent in olaparib resistant tumors. Additional PDX models would have been good to see; however, extensive data in appropriate cell line models as well as in silico data analyses make a strong case for the claims of these authors. The following points should be addressed:

1. How are the authors correlating dose ranges between 38 – 285 nM CX-5461 with plasma

concentrations of 584.1 nM to 3.3 μ M (Figs. 1D & E)?

2. The comment is made that functional defects in the HR pathway correlate with CX-5461 sensitivity. OVCAR-3 cells are shown as the most sensitive (Figure 1A) but do not have defective HR signalling. Can the authors please comment on (correct?) this?

3. Can TP53 mutations be separated into "type", or at the least missense versus other mutations, to exclude any correlation between TP53 mutation type and CX-5461 sensitivity?

4. p53 is a negative regulator of MYC. The authors have identified a correlation with a MYC gene signature but not with TP53 mutation status when it comes to CX-5461 expression. Can they please comment on this point?

Response to reviewers; Sanij et al
Reviewers' comments highlighted in BLUE:

Reviewer #1:

We thank the reviewer for their positive and constructive comments:... *"In general terms, the cancer therapy-related aspects of the manuscript are very solid, but the authors are too ambitious in their explanation of the molecular bases of CX-5461 action, which seem too complex to be satisfactorily explained with the results presented. Thus, some of the claims would need to be further substantiated, or alternatively, presented in a more descriptive fashion."* We have performed a number of additional experiments to further define the mechanisms of CX-5461 action to address the reviewer's concerns (details below) and we have also simplified data interpretation to be more descriptive as suggested by the reviewer.

"1. Regarding the observed accumulation of R-loops. It is not easy for me to understand how an inhibition of transcription initiation, as caused by CX-5461, can generate an increase in R-loops. Authors propose that the absence of Pol I at the promoter favors R-loop formation, but in most models the formation of the R-loop is co-transcriptional, and would therefore be reduced upon transcription inhibition." This is an excellent point. In the revised manuscript, we clarify that CX-5461 mediates stabilization of pre-existing R-loops, as opposed to de novo R-loops formation in OVCAR8 cells. Indeed, evidence of R-loops stabilization upon inhibition of Pol I transcription was recently reported following mild hypo-osmotic stress (Velichko et al., NAR May 2019).

R-loops occur naturally at the rRNA genes as Pol I transcription is highly dynamic and more intense than Pol II transcription. Transcription initiation at ribosomal RNA (rRNA) gene promoters occurs every ~1.4 sec and approximately 100 ± 29 polymerases molecules transcribe the 13 kb single rRNA gene (French et al., MCB 2003; Ingrid Grummt G&D 2003). Inhibition of transcription initiation by CX-5461 will inhibit formation of new R-loops. However, as transcription initiation is tightly coupled to processing, inhibition of initiation can coincide with inhibition of processing of precursor rRNAs leading to stabilization of R-loops from transcripts already generated by elongating Pol I molecules.

We have replaced our description of "R-loops formation" to "R-loops stabilisation" in the revised manuscript and included a clarification of this observation in OVCAR8 cells on page 10, 2nd paragraph.

"As a matter of fact, in HR-deficient cell lines, CX-5461 causes a reduction R-loops (Figure 3D), which the authors fail to discuss appropriately. Furthermore, there is no correlation between the levels of R-loops and replication stress markers cell death. Therefore, the involvement of R-loop accumulation in CX-5461 toxicity, and the connection with replication stress that authors claim does not seem sufficiently substantiated. This could be done, for example, by analyzing a possible suppression of the replication stress and cell death-phenotypes upon RNase H overexpression." As the reviewer rightly points out, in HR-deficient cells the involvement of R-loop accumulation did not correlate with CX-

5461 toxicity. We thank the reviewer for their suggestion and have carried out the suggested RNase H overexpression experiment (Supplementary Figure 7 and Supplementary Figure 9D-E). Indeed, the new data show that RNase H overexpression reduced nucleolar R-loops levels in OVCAR RAD51C KO cells treated with CX-5461 compared to vehicle control, however this did not prevent CX-5461-mediated nucleolar localisation of phosphorylated RPA at S33 within 3h of treatment indicating the presence of ssDNA and fork stalling (Supplementary Figure 7B&C). At 24 hours post CX-5461 treatment, RNase H overexpression partially reduced global replication stress marked by phosphorylation of RPA32 S4/S8 phosphorylation however this did not rescue CX-5461 growth inhibitory effects in OVCAR8 RAD51C KO and in patient-derived WEHICS62 cells (Supplementary Figure 7D&E and Supplementary Figure 9D-E). Altogether, our data implicates CX-5461 in inducing ssDNA structures and replication stress at rDNA, with stabilization of R-loops being indicative of chromatin defects at rDNA and contributing to CX-5461-mediated DDR but not being essential for CX-5461 efficacy.

We have added the description of this data in the revised manuscript on page 11, 3rd paragraph and page 16, 2nd paragraph.

“2. The activation of ATR by CX-5461 treatment in non-replicating cells is not explained by a model that is dependent on replication stress and fork stability. This should be, at least, discussed in more detail. In addition, it could be useful to determine whether DNA replication is required for CX-5461 toxicity. If this were the case, the effects observed on non-replicating cells would be irrelevant from a therapeutic perspective and could be obviated.”

We thank the reviewer for this suggestion. In the revised manuscript, we have expanded on our model of activation of ATR in HR-deficient cells, which we observed in both replicating and non-replication cells. HR-deficient RAD51C KO OVCAR8 cells exhibited high levels of nucleolar activation of ATR (Figure 4B) and γ H2AX foci (Figure 5A&B) in both the EdU-positive and -negative populations compared to CX-5461-treated HR-proficient cells. As noted by the reviewer, this data suggests that replication stress is not the only cause of CX-5461-induced DNA damage. We note that while, CX-5461 mediates replication stress and DNA damage in HR-proficient cells, DNA damage is exacerbated and occurs independent of the cell cycle stage due to the lack of repair in HR-deficient cells and have included this discussion on page 12, 2nd paragraph.

In order to determine whether DNA replication is required for CX-5461 toxicity, we assessed the consequences of activating nucleolar DDR and DNA damage in non-replicating cells, utilising the FUCCI system to analyse the effect of CX-5461 treatment on G1, S and G2 isolated FUCCI-labelled OVCAR8 and RAD51C KO OVCAR8 cells (Supplementary Figure 6B). Regardless of the cell cycle stage, CX-5461 treatment led to G2 cell cycle arrest. Due to the lack of p53 activity in OVCAR8 cells, the G2 checkpoint arrest appears to be the major response to CX-5461 thus suggesting DNA replication is required for CX-5461 growth inhibitory effects. This data is presented on page 11, 1st paragraph.

“3. Regarding the DNA fiber analysis of DNA replication. This technique measures global DNA replication, and not specifically at the rDNA. The effects observed are therefore unlikely to be directly caused by problems locally occurring at the rDNA as a consequence of inhibiting transcription by CX-

5461. Authors need to address this important limitation; otherwise, the conclusions drawn from the analysis of replication are seriously compromised.”

We agree with the reviewer that clarification on the connection between rDNA replication stress and the global replication stress signal by fibre analysis is important. We have now included in the revised manuscript an additional subsection in the discussion (page 19, 2nd paragraph) to highlight our findings in the context of nucleolar-specific DDR being a major effector in cellular stress response.

We now discuss how CX-5461-mediated nucleolar-specific DDR initiates a cellular stress response by arresting cell cycle progression, leading to stalling of replication and destabilization of replication forks via MRE11 activity leading to DNA damage (Figure 2E, Figure 5C, Figure 6 and Figure 8D-F). We note that this finding is consistent with the nucleolus being a central hub for coordinating the cellular stress response. For example, work from numerous laboratories demonstrated nucleolar stress signalling relies on the dynamic sequestration and release of nucleolar factors involved in replication and transcription, providing a link between ribosome biogenesis, cell-cycle progression, and stress signalling (reviewed in Boulon et al., Mol Cell 2010).

“In addition, an analysis of replication-fork asymmetry would be more appropriate to unambiguously claim an effect on replication-fork stability. Finally, I fail to see how the results presented demonstrate an effect of CX-5461 in overcoming fork protection in olaparib-resistant cell lines. This should be directly addressed by, for example, checking the effect of CX-5461 on fork stability in HU-treated WEHICS62 cells.”

We thank reviewer for their suggestion. We have performed new DNA fibre assays to measure replication-fork stability by analysing fork asymmetry (New Figure 8D) that demonstrate CX-5461 treatment destabilizes forks and the combination with HU enhances fork instability, page 16, 2nd paragraph.

Specific comments:

“4. For the analysis of CX-5461 sensitivity in HR-depleted cells (Figure 1H), the absolute numbers, and not just the combination index should be presented.” We have included in Figure 1H a graph of absolute cell numbers demonstrating CX-5461 cooperates with knockdown of HR genes in inhibiting OVCAR4 cell growth.

“5. The claimed G2 accumulation in response to CX-5461 treatment (Figure 2B) is difficult to perceive in the way the data are presented.” We have included in the revised manuscript new cell cycle analysis using the FUCCI system (Figure 2C&D) that confirms CX-5461 activates G2M cell cycle arrest in OVCAR8 and RAD51C KO OVCAR8 cells.

“6. The quality of the Western blot presented in Figure 2C is not sufficient. In addition, the effects claimed are not easily observed. For example, the accumulation of pCHK1/2 and γ H2AX.”

We have included a new set of western blot analyses of the DDR in OVCAR8 and OVCAR8 RAD51C KO following 6h and 24h treatment with CX-5461 (New Figure 2E). Consistent with previous data, CX-5461 induced ATM/ATR signalling, pCHK1/2 and low induction in γ H2AX levels in both cell lines. Strikingly, greater increases in S4/S8 phosphorylation of RPA32, which protects ssDNA and marks

persistent replication stress, were observed in HR-deficient cells following 100 nM and 1 μ M CX-5461 compared to HR proficient OVCAR8 cells.

“7. The accumulation of γ H2AX and RAD51 foci in response to CX-5461, BMN-673 and their combination should also be analyzed in HR-deficient cells.” We have performed this additional experiment as the reviewer suggested. We replaced γ H2AX IF staining in HR-proficient and HR-deficient cells in old Figure 5C and γ H2AX IF and RAD51 IF data in OVCAR8 cells in old Figure 6A-B with new data in New Figure 6A-B showing the combination of CX-5461 and BMN-673 leads to a significant increase in γ H2AX foci and RAD51 foci formation in HR-proficient and HR-deficient cells suggesting their cooperation in exacerbating DNA damage, page 13, 1st paragraph.

Reviewer #2

“This is an interesting study evaluating the impact of the RNA polymerase I inhibitor CX-5461 in high grade serous ovarian cancer (HGSOC).... Overall, the studies appear carefully performed and are clearly presented. The results have broad implications for understanding how DDR inhibitors could be combined, how PARP inhibitors affect DDR, and the essential role for replication for instability in sensitivity to these inhibitors.” We thank the reviewer for their positive comments.

“The key weakness in this paper that makes the manuscript difficult for this reviewer to easily understand is a failure to connect rDNA replication stress to a more global replication stress signal. Given the relatively small representation of rDNA to the overall size of the genome, it is not clear if the authors are arguing that this small region of replication stress is causing the over-arching effects on DDR signaling, cell death and DNA synthesis (CldU/IdU assays) of the RNA PolI inhibitor. Many of these assays will assess both ribosomal and non-ribosomal DNA regions. Thus, are there other mechanisms beyond effects on rRNA synthesis and rDNA replication stress that contribute to the cytotoxicity effects of the drug or are there off-target effects of the drug that also influence the efficacy.”

Reviewer 1 also asked for clarification of the connection between rDNA replication stress and the global replication stress signal. As noted above, we have included in the revised manuscript an additional subsection in the discussion (page 19, 2nd paragraph) to highlight our findings in the context of nucleolar-specific DDR as a major effector in cellular stress response.

Regarding additional mechanisms of response to CX-5461, we now reference (page 19, 2nd paragraph) new work that has recently appeared in BioRxiv showing the sensitivity profile of CX-5461 to resemble a topoisomerase II (TOP2) inhibitor (Olivieri et al., BioRxiv 2019). TOP 2a is an essential component of the Pol I pre-initiation complex and while our data clearly demonstrate CX-5461 inhibits Pol I transcription and activates nucleolar specific DDR, it is plausible that it does so by trapping TOP2 at rDNA and this perhaps influences TOP2 activity across the genome. We have included this discussion in the revised manuscript (page 19, 2nd paragraph). While the issue of off-target effect is important for characterisation of mechanism of action, our extensive data clearly

demonstrate CX-5461 induces nucleolar DDR and the net effects in the destabilization of replication forks have important clinical implications as discussed on page 19, 3rd paragraph.

“The only significant issue identified in the data presentation and discussion is Figure 5A EdU staining of CX-5461 treatment – the EdU staining that outlines two nuclei does not seem to match the DAPI stained nuclei, suggesting this image is not of the same microscopic field as the other two images shown (DAPI and H2AX) for this sub-figure. This should be addressed and/or corrected.” We most sincerely thank the reviewer for this comment and we have updated Figure 5A to include the correct EdU image.

Reviewer #3

“Sanij and colleagues have presented a comprehensive and interesting paper investigating the RNA polymerase I inhibitor, CX-5461, including how it enhances the effect of PARP inhibitors. It is stated (earlier publications) that CX-5461 has shown efficacy in both p53 wild-type and mutant malignancies, functioning as a p53 independent activator of the DNA damage response. The data presented in this paper supports this as there was no statistical difference between wt or mutant p53 cell lines in response to CX-5461.

RNA pol I inhibition has also been previously published to show synthetic lethality with BRCA mutant cells. The in vivo work presented, including that CX-5461 and the PARPi olaparib given together was well tolerated and had additional effect over olaparib alone is promising. Further, that CX-5461 may have benefit when given as a single agent in olaparib resistant tumors. Additional PDX models would have been good to see; however, extensive data in appropriate cell line models as well as in silico data analyses make a strong case for the claims of these authors.”

We thank the reviewer for their encouraging comments

“The following points should be addressed:

1. How are the authors correlating dose ranges between 38 – 285 nM CX-5461 with plasma concentrations of 584.1 nM to 3.3 μM (Figs. 1D & E)?”

We have modified the above sentence to clarify our point, page 7, the end of the first paragraph to “The data shows CX-5461 is on-target in inhibiting Pol I transcription at doses 10-fold less than the plasma concentrations range (584.1 nM to 3.3 μM) used in the Phase I CX-5461 dose escalation study.”

“2. The comment is made that functional defects in the HR pathway correlate with CX-5461 sensitivity. OVCAR-3 cells are shown as the most sensitive (Figure 1A) but do not have defective HR signalling. Can the authors please comment on (correct?) this?”

The reviewer is correct in pointing out OVCAR3 cells are HR-proficient. In addition to HR deficiency, sensitivity to CX-5461 correlates with high basal level of Pol I transcription (Figure 1C). The OVCAR3 cell line was included in this analysis. We have emphasised this point in page 7, last line.

“3. Can TP53 mutations be separated into “type”, or at the least missense versus other mutations, to exclude any correlation between TP53 mutation type and CX-5461 sensitivity?” In Supplementary

Table 1, we provide the TP53 mutation status of the ovarian cancer cell lines. Only 2 cell lines exhibited deletion “type” of p53 mutations as opposed to missense mutations. While the reviewer raised an interesting point, we are unable to do this correlation.

“4. p53 is a negative regulator of MYC. The authors have identified a correlation with a MYC gene signature but not with TP53 mutation status when it comes to CX-5461 expression. Can they please comment on this point?”

We believe that the regulatory effects of p53 on MYC expression have only a minor effect on the activity of MYC in driving Pol I transcription at multiple levels (Poortinga et al., NAR 2012). We have demonstrated MYC upregulation of Pol I transcription is required to drive malignant transformation in the Eμ-MYC lymphoma model (Chan et al., Science signalling 2011; Bywater et al., Cancer Cell 2012). Hence, MYC activation is a much stronger predictor of response than p53 status. We have added a comment to address reviewer’s question to the discussion on page 8, 1st paragraph.

Reviewers' comments:

Reviewer #1 (Remarks to the Author); expert on DNA Damage:

I appreciate the effort of the authors in addressing my comments. The work has improved, and some aspects are now clearer. All my experimental suggestions have been included, and, in this sense, I just have one additional concern. The Western blot in Figure 2E still shows strong differences in the total amount of proteins such as ATR, ATM, CHK1, CHK2 and RPA. I believe that this is due to technical problems that should be solved, as they otherwise invalidate the conclusions that can be drawn regarding differences in DDR activation.

I still have important concerns regarding the conclusions drawn from the result, specifically in the more mechanistic DDR-related aspects. I fail to see any evidence indicating a specific effect in nucleoli, and this is purely based on the fact that CX-5461 is a reported RNA Pol I inhibitor. You can clearly observe that in response to the drug pATR, pRPA and gH2AX accumulate all over the nucleus, and only very partially in nucleoli. I think it is misleading to just quantify this signal and conclude that CX-5461 is inducing nucleolar stress. R-loops do seem to be specific for nucleoli, but this can be related to the high abundance of RNA in these regions, and can also be observed with other DNA-damaging agents. In any case, as authors appropriately discuss, R-loop accumulation does not seem to be the cause of the toxicity. Results could be very well explained in the context of CX-5461 being a more general DNA-damaging agent, which, as the authors say, fits with recent results presented in a preprint manuscript (Olivieri 2019, BioRxiv). Furthermore, I also fail to see a qualitative difference between the behavior of pATR, pRPA and gH2AX regarding S-phase- and/or HRD-dependency. In all cases CX-5461 causes an increase (which as mentioned above is global and not nucleoli-specific) that is further increased by replication and even more upon HRD. Quantitatively the responses are different, perhaps due to differences in signal-to-noise ratios of the given antibody, so in some cases the differences are not as strong and not reaching statistical significance, but the trend can be observed. Altogether, this would suggest a simpler model that would allow authors to put a focus on the implications for therapy, which seem more solid, regardless of the mechanism of action of CX-5461.

Finally, the general organization of the manuscript, and some specific parts in particular, are very confusing and difficult to follow. Thus, authors should make an effort in streamlining the text and improving its flow, with a more rational order of text and figures. For example, the text corresponding to Figure 2 is not well constructed and is very difficult to follow. In many cases, not sufficient experimental detail is provided in the text in order to properly understand the figures. Also, the last part of the manuscript (corresponding to Figure 9) is directly connected and actually presented as a validation of the results presented in Figure 1F. Going back and forth from the molecular mechanisms to the therapy related aspects is somewhat confusing and should be avoided.

Specific comments:

- Is the y axis of Fig1E mislabeled?
- I do not agree that an increase in 8N content indicates replication failure, but rather a segregation problem (which would be consistent with TOP2 problems, by the way).
- Would move Fig2C to supplementary, and make more clear the difference between Fig2B and 2D in the figure, so it can be directly seen without checking the legend.
- In the experiment in which the FUCCI cells are sorted, this should be indicated in the text. Otherwise it is difficult to understand how the experiment is made.

Reviewer #2 (Remarks to the Author); expert on mouse models and PARPi resistance:

The authors have adequately addressed this reviewer's concerns

Reviewer #3 (Remarks to the Author); expert on translation:

The authors have adequately addressed all points raised.

Having satisfied Reviewers 2 and 3, we have revised our manuscript as suggested by Reviewer 1 to focus on the implications for CX-5461 therapy and toned down our conclusions regarding nucleolar stress as a mechanism for activation of DDR. Please note the changes in the discussion on pages 18 and 19.

Our response to Reviewer 1's comments is highlighted in blue. List of changes to figures include:

- Figure 9A is now Figure 1H
- Figure 2C (FUCCI data) is now Supplementary Figure 3E
- Quantitation of total pATR T1989 signal intensity added to Figure 4B
- Quantitation of total pRPA S33 signal intensity added to Figure 4D
- Modifications to schematic in Figure 6D
- Quantitation of total pATR T1989 signal intensity added to Figure 8E
- Quantitation of total pRPA S33 signal intensity added to Supplementary Figure 7C

Reviewer #1:

I appreciate the effort of the authors in addressing my comments. The work has improved, and some aspects are now clearer. All my experimental suggestions have been included, and, in this sense, I just have one additional concern. We thank the reviewer for their encouraging comment.

The Western blot in Figure 2E still shows strong differences in the total amount of proteins such as ATR, ATM, CHK1, CHK2 and RPA. I believe that this is due to technical problems that should be solved, as they otherwise invalidate the conclusions that can be drawn regarding differences in DDR activation. While there are differences in total ATR, ATM, CHK1, CHK2 and RPA32 between the 6-hour and the 24-hour timepoints, we believe this does not invalidate our conclusions of increased DDR signalling in CX-5461 treated samples at both time points. When compared to corresponding vehicle controls there are clear increases in the phosphorylated forms at both time points. The western blots we provided are representative of three biological repeats. Here, we provide an additional western blot data of time course treatment of CX-5461 of the OVCAR8 HR-proficient and OVCAR8 RAD51C KO HR-deficient OVCAR8 cell lines again showing that changes in total protein level do occur but importantly, compared to vehicle controls at each timepoint, there are again clear increases in the levels of phosphorylated forms in response to treatment. We appreciate the reviewer's comment on quantitative responses being different between the two cell lines, perhaps due to differences in signal-to-noise ratios of the given antibody. Therefore, in the revised text we deleted a sentence on page 9 (second paragraph) to keep the text focused on CX-5461 activating DDR signalling in both cell lines but retained our statement "greater increases in S4/S8 phosphorylation of RPA32 were observed in HR-deficient cells following 100 nM and 1 \$\mu\$ M CX-5461 compared to HR proficient OVCAR8 cells". Under these conditions (in n=5 repeats), despite reduced signal of total RPA32 levels following 24-hour treatment with CX-5461, we observed an increase in the levels of the phosphorylated forms. A reduced antibody affinity for the phosphorylated form of proteins compared to unphosphorylated forms is commonly observed and could explain some of the observed changes in total protein. We believe that this extensive analysis of multiple readouts including the appropriate controls fully justifies our revised conclusions.

I still have important concerns regarding the conclusions drawn from the result, specifically in the more mechanistic DDR-related aspects. I fail to see any evidence indicating a specific effect in nucleoli, and this is purely based on the fact that CX-5461 is a reported RNA Pol I inhibitor. You can clearly observe that in response to the drug pATR, pRPA and γH2AX accumulate all over the nucleus, and only very partially in nucleoli. I think it is misleading to just quantify this signal and conclude that CX-5461 is inducing nucleolar stress. We have moderated our conclusions regarding nucleolar stress as a mechanism for activation of DDR and made relevant changes to the schematic in Figure 6D including adding the quantitation of total nuclear signal as separate graphs to the quantitation of nucleolar pATR, pRPA signal in Fig 4, Fig 8E and Suppl Fig 7C. Please note quantitation of γH2AX foci counts in Fig 5B and Fig 6B has not changed as it already measures total γH2AX foci counts. We have toned down our references to nucleolar-specific DDR throughout the manuscript in view of the changes in total nuclear DDR.

R-loops do seem to be specific for nucleoli, but this can be related to the high abundance of RNA in these regions, and can also be observed with other DNA-damaging agents. In any case, as authors appropriately discuss, R-loop accumulation does not seem to be the cause of the toxicity. Results could be very well explained in the context of CX-5461 being a more general DNA-damaging agent, which, as the authors say, fits with recent results presented in a preprint manuscript (Olivieri 2019, BioRxiv). We have referenced the work by Olivieri et al and a new report by (Bruno et al., PNAS 2020) in the introduction (page 6) and focused our discussion on the therapeutic potential of CX-5461 in the context of a potential TOP II poison (page 19 1st paragraph).

Furthermore, I also fail to see a qualitative difference between the behavior of pATR, pRPA and γH2AX regarding S-phase- and/or HRD-dependency. In all cases CX-5461 causes an increase (which as mentioned above is global and not nucleoli-specific) that is further increased by replication and even more upon HRD. Quantitatively the responses are different, perhaps due to differences in signal-to-noise ratios of the given antibody, so in some cases the differences are not as strong and not reaching statistical significance, but the trend can be observed. Altogether, this would suggest a simpler model that would allow authors to put a focus on the implications for therapy, which seem more solid, regardless of the mechanism of action of CX-5461. We thank the reviewer for their constructive comment. To allow us to focus on a more simple model of the induction of the DDR, we have removed the statistical analyses of qualitative differences between CX-5461 induced pATR, pRPA and γH2AX regarding S-phase- and/or HRD-dependency in Figure 4 and Figure 5B. We have modified the text on pages 11 and 12 to discuss that CX-5461 induces replication stress and DNA

damage with less emphasis on quantitative comparisons between the different condition the reviewer outlined above. In the revised text we provide a simpler description of the data that CX-5461-mediated phosphorylation of RPA and ATR was independent of the cell cycle stage and was not restricted to the nucleoli (page 11).

Finally, the general organization of the manuscript, and some specific parts in particular, are very confusing and difficult to follow. Thus, authors should make an effort in streamlining the text and improving its flow, with a more rational order of text and figures. For example, the text corresponding to Figure 2 is not well constructed and is very difficult to follow. In many cases, not sufficient experimental detail is provided in the text in order to properly understand the figures. Also, the last part of the manuscript (corresponding to Figure 9) is directly connected and actually presented as a validation of the results presented in Figure 1F. Going back and forth from the molecular mechanisms to the therapy related aspects is somewhat confusing and should be avoided. We have streamlined the text describing Figure 2 and deleted two sentences on pages 9 and 15 to improve the flow of the text as suggested. We have moved data in Figure 9A to Figure 1. We have modified the text to focus on the therapy related aspect of the results as indicated in the tracked changes.

We have also made the changes suggested below under “specific comments”

Specific comments:

- Is the y axis of Fig1E mislabeled? We have clarified the label in Figure1E. Y axis now displays CX-5461 IC50 (nM) for Pol I transcription inhibition

- I do not agree that an increase in 8N content indicates replication failure, but rather a segregation problem (which would be consistent with TOP2 problems, by the way). We thank the reviewer for their insight. We have included “...CX-5461 induces defects in chromosome segregation possibly due to persistent DNA replication stress and DNA damage in mitosis (Blackford and Stucki, 2020 Trends in Biochemical Sciences), page 9 1st paragraph.

- Would move Fig2C to supplementary, and make more clear the difference between Fig2B and 2D in the figure, so it can be directly seen without checking the legend. We have made this change and moved Figure 2C to Supplementary Figure 3E.

- In the experiment in which the FUCCI cells are sorted, this should be indicated in the text. Otherwise it is difficult to understand how the experiment is made. We have added additional text to clarify experimental details on pages 11 and 12 as the reviewer suggested.

REVIEWERS' COMMENTS:

Reviewer #1 (Remarks to the Author):

The authors have now addressed all my concerns. The manuscripts can now be accepted, pending careful proofreading, as there are still abundant typos and errors.

Here, we provide response to Reviewer 1's comment.

Reviewer 1: "The authors have now addressed all my concerns. The manuscripts can now be accepted, pending careful proofreading, as there are still abundant typos and errors."

We thank Reviewer 1 for their encouraging comment. We have edited our manuscript to avoid typing errors as they suggested.